# Self-Supervised Learning
# with Kernel Dependence Maximization

**Yazhe Li**[*]
DeepMind and Gatsby Unit, UCL
yazhe@google.com

**Roman Pogodin**[*]
Gatsby Unit, UCL
roman.pogodin.17@ucl.ac.uk

**Danica J. Sutherland**
UBC and Amii[†]
dsuth@cs.ubc.ca

**Arthur Gretton**
Gatsby Unit, UCL
arthur.gretton@gmail.com

## Abstract

We approach self-supervised learning of image representations from a statistical dependence perspective, proposing Self-Supervised Learning with the Hilbert-Schmidt Independence Criterion (SSL-HSIC). SSL-HSIC maximizes dependence between representations of transformations of an image and the image identity, while minimizing the kernelized variance of those representations. This framework yields a new understanding of InfoNCE, a variational lower bound on the mutual information (MI) between different transformations. While the MI itself is known to have pathologies which can result in learning meaningless representations, its bound is much better behaved: we show that it implicitly approximates SSL-HSIC (with a slightly different regularizer). Our approach also gives us insight into BYOL, a negative-free SSL method, since SSL-HSIC similarly learns local neighborhoods of samples. SSL-HSIC allows us to directly optimize statistical dependence in time linear in the batch size, without restrictive data assumptions or indirect mutual information estimators. Trained with or without a target network, SSL-HSIC matches the current state-of-the-art for standard linear evaluation on ImageNet [1], semi-supervised learning and transfer to other classification and vision tasks such as semantic segmentation, depth estimation and object recognition. Code is available at https://github.com/deepmind/ssl_hsic.

## 1 Introduction

Learning general-purpose visual representations without human supervision is a long-standing goal of machine learning. Specifically, we wish to find a feature extractor that captures the image semantics of a large unlabeled collection of images, so that e.g. various image understanding tasks can be achieved with simple linear models. One approach takes the latent representation of a likelihood-based generative model [2–8]; such models, though, solve a harder problem than necessary since semantic features need not capture low-level details of the input. Another option is to train a *self-supervised* model for a "pretext task," such as predicting the position of image patches, identifying rotations, or image inpainting [9–14]. Designing good pretext tasks, however, is a subtle art, with little theoretical guidance available. Recently, a class of models based on contrastive learning [15–22] has seen substantial success: dataset images are cropped, rotated, color shifted, etc. into several *views*, and features are then trained to pull together representations of the "positive" pairs of views of the same

---

[*]These authors contributed equally.

[†]Work done in part while at the Toyota Technological Institute at Chicago.

35th Conference on Neural Information Processing Systems (NeurIPS 2021).

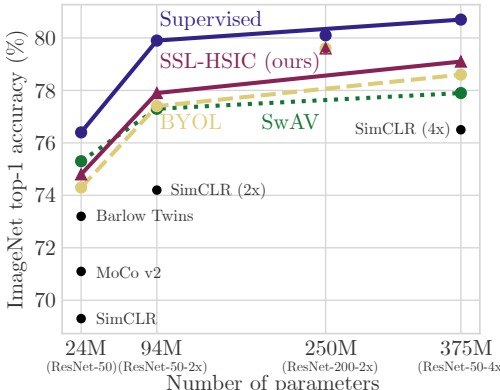

Figure 1: Top-1 accuracies with linear evaluation for different ResNet architecture and methods: supervised (as in [25]), SSL-HSIC (with a target network; ours), BYOL [25], SwAV [18], SimCLR [19], MoCo v2 [20] and Barlow Twins [22].

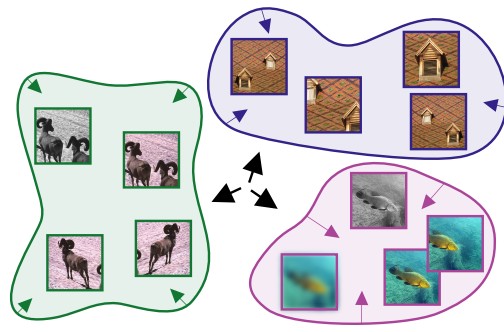

Figure 2: Statistical dependence view of contrastive learning: representations of transformed images should highly depend on image identity. Measuring dependence with HSIC, this pushes different images' representation distributions apart (black arrows) and pulls representations of the same image together (colored shapes).

source image, and push apart those of "negative" pairs (from different images). These methods are either understood from an information theoretic perspective as estimating the mutual information between the "positives" [15], or explained as aligning features subject to a uniformity constraint [23]. Another line of research [24, 25] attempts to learn representation without the "negative" pairs, but requires either a target network or stop-gradient operation to avoid collapsing.

We examine the contrastive framework from a statistical dependence point of view: feature representations for a given transformed image should be highly dependent on the image identity (Figure 2). To measure dependence, we turn to the Hilbert-Schmidt Independence Criterion (HSIC) [26], and propose a new loss for self-supervised learning which we call SSL-HSIC. Our loss is inspired by HSIC Bottleneck [27, 28], an alternative to Information Bottleneck [29], where we use the image identity as the label, but change the regularization term.

Through the dependence maximization perspective, we present a unified view of various self-supervised losses. Previous work [30] has shown that the success of InfoNCE cannot be solely attributed to properties of mutual information, in particular because mutual information (unlike kernel measures of dependence) has no notion of geometry in feature space: for instance, *all* invertible encoders achieve maximal mutual information, but they can output dramatically different representations with very different downstream performance [30]. Variational bounds on mutual information do impart notions of locality that allow them to succeed in practice, departing from the mutual information quantity that they try to estimate. We prove that InfoNCE, a popular such bound, in fact approximates SSL-HSIC with a variance-based regularization. Thus, InfoNCE can be thought of as working because it implicitly estimates a kernel-based notion of dependence. We additionally show SSL-HSIC is related to metric learning, where the features learn to align to the structure induced by the self-supervised labels. This perspective is closely related to the objective of BYOL [25], and can explain properties such as alignment and uniformity [23] observed in contrastive learning.

Our perspective brings additional advantages, in computation and in simplicity of the algorithm, compared with existing approaches. Unlike the indirect variational bounds on mutual information [15, 31, 32], SSL-HSIC can be directly estimated from mini-batches of data. Unlike "negative-free" methods, the SSL-HSIC loss itself penalizes trivial solutions, so techniques such as target networks are not needed for reasonable outcomes. Using a target network does improve the performance of our method, however, suggesting target networks have other advantages that are not yet well understood. Finally, we employ random Fourier features [33] in our implementation, resulting in cost linear in batch size.

Our main contributions are as follows:

- We introduce SSL-HSIC, a principled self-supervised loss using kernel dependence maximization.
- We present a unified view of contrastive learning through dependence maximization, by establishing relationships between SSL-HSIC, InfoNCE, and metric learning.

- Our method achieves top-1 accuracy of 74.8% and top-5 accuracy of 92.2% with linear evaluations (see Figure 1 for a comparison with other methods), top-1 accuracy of 80.2% and Top-5 accuracy of 94.7% with fine-tuning, and competitive performance on a diverse set of downstream tasks.

## 2 Background

### 2.1 Self-supervised learning

Recent developments in self-supervised learning, such as contrastive learning, try to ensure that features of two random views of an image are more associated with each other than with random views of other images. Typically, this is done through some variant of a classification loss, with one "positive" pair and many "negatives." Other methods can learn solely from "positive" pairs, however. There have been many variations of this general framework in the past few years.

Van den Oord et al. [15] first formulated the InfoNCE loss, which estimates a lower bound of the mutual information between the feature and the context. SimCLR [19, 34] carefully investigates the contribution of different data augmentations, and scales up the training batch size to include more negative examples. MoCo [17] increases the number of negative examples by using a memory bank. BYOL [25] learns solely on positive image pairs, training so that representations of one view match that of the other under a moving average of the featurizer. Instead of the moving average, SimSiam [24] suggests a stop-gradient on one of the encoders is enough to prevent BYOL from finding trivial solutions. SwAV [18] clusters the representation online, and uses distance from the cluster centers rather than computing pairwise distances of the data. Barlow Twins [22] uses an objective related to the cross-correlation matrix of the two views, motivated by redundancy reduction. It is perhaps the most related to our work in the literature (and their covariance matrix can be connected to HSIC [35]), but our method measures dependency more directly. While Barlow Twins decorrelates components of final representations, we maximize the dependence between the image's abstract identity and its transformations.

On the theory side, InfoNCE is proposed as a variational bound on Mutual Information between the representation of two views of the same image [15, 32]. Tschannen et al. [30] observe that InfoNCE performance cannot be explained solely by the properties of the mutual information, however, but is influenced more by other factors, such as the formulation of the estimator and the architecture of the feature extractor. Essentially, representations with the same MI can have drastically different representational qualities.

To see this, consider a problem with two inputs, $A$ and $B$ (Figure 3, green and purple), and a one-dimensional featurizer, parameterized by the integer $M$, which maps $A$ to $\text{Uniform}(\{0, 2, \ldots, 2M\})$ and $B$ to $\text{Uniform}(\{1, 3, \ldots, 2M + 1\})$. When $M = 0$, the inputs are encoded into linearly separable features $A = 0$ and $B = 1$ (Figure 3, bottom). Otherwise when $M > 0$, they are interspersed like $ABABABAB$ – a representation which is much harder to work with for downstream learners. Nevertheless, the mutual information between the features of any two augmentations of the same input (a positive pair) is independent of $M$, that is $H[Z_1] - H[Z_1|Z_2] = \log 2$ for any $M$. Note that InfoNCE would strongly prefer $M = 0$, indeed behaving very differently from MI.

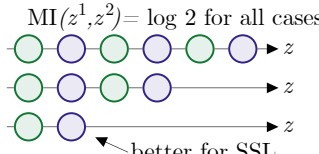

$\text{MI}(z^1, z^2) = \log 2$ for all cases

better for SSL

Figure 3: Three distributions of positive examples for two classes (green and purple) that have the same mutual information, but drastically different quality for downstream learners.

Later theories suggest that contrastive losses balance alignment of individual features and uniformity of the feature distribution [23], or in general alignment and some loss-defined distribution [36]. We propose to interpret the contrastive loss through the lens of statistical dependence, and relate it to metric learning, which naturally leads to alignment and uniformity.

### 2.2 Hilbert-Schmidt Independence Criterion (HSIC)

The Hilbert-Schmidt Independence Criterion (HSIC) [26] is a kernel-based measure of dependence between probability distributions. Like mutual information, for a wide range of kernels $\text{HSIC}(X, Y) = 0$ if and only if $X$ and $Y$ are independent [37], and large values of the measure correspond to "more dependence." Unlike mutual information, HSIC incorporates a notion of ge-

ometry (via the kernel choice), and is both statistically and computationally easy to estimate. It has been used in a variety of applications, particularly for independence testing [38], but it has also been maximized in applications such as feature selection [39], clustering [40, 41], active learning [42], and as a classification loss called HSIC Bottleneck [27, 28] (similar ideas were expressed in [43, 44]).

HSIC measures the dependence between two random variables by first taking a nonlinear feature transformation of each, say $\phi : \mathcal{X} \to \mathcal{F}$ and $\psi : \mathcal{Y} \to \mathcal{G}$ (with $\mathcal{F}$ and $\mathcal{G}$ reproducing kernel Hilbert spaces, RKHSes[1]), and then evaluating the norm of the cross-covariance between those features:

$$\mathrm{HSIC}(X, Y) = \left\| \mathbb{E}[\phi(X)\, \psi(Y)^\top] - \mathbb{E}[\phi(X)]\, \mathbb{E}[\psi(Y)]^\top \right\|_{HS}^2. \tag{1}$$

Here $\|\cdot\|_{HS}$ is the Hilbert-Schmidt norm, which in finite dimensions is the usual Frobenius norm. HSIC measures the scale of the correlation in these nonlinear features, which allows it to identify nonlinear dependencies between $X$ and $Y$ with appropriate features $\phi$ and $\psi$.

Inner products in an RKHS are by definition *kernel functions*: $k(x, x') = \langle \phi(x), \phi(x') \rangle_{\mathcal{F}}$ and $l(y, y') = \langle \psi(y), \psi(y') \rangle_{\mathcal{G}}$. Let $(X', Y'), (X'', Y'')$ be independent copies of $(X, Y)$; this gives

$$\mathrm{HSIC}(X, Y) = \mathbb{E}\left[ k(X, X') l(Y, Y') \right] - 2 \mathbb{E}\left[ k(X, X') l(Y, Y'') \right] + \mathbb{E}\left[ k(X, X') \right] \mathbb{E}\left[ l(Y, Y') \right]. \tag{2}$$

HSIC is also straightforward to estimate: given i.i.d. samples $\{(x_1, y_1), \ldots, (x_N, y_N)\}$ drawn i.i.d. from the joint distribution of $(X, Y)$, Gretton et al. [26] propose an estimator

$$\widehat{\mathrm{HSIC}}(X, Y) = \frac{1}{(N-1)^2} \mathrm{Tr}(KHLH), \tag{3}$$

where $K_{ij} = k(x_i, x_j)$ and $L_{ij} = l(y_i, y_j)$ are the kernel matrices, and $H = I - \frac{1}{N} \mathbf{1}\mathbf{1}^\top$ is called the centering matrix. This estimator has an $O(1/N)$ bias, which is not a concern for our uses; however, an unbiased estimator with the same computational cost is available [39].

## 3   Self-supervised learning with Kernel Dependence Maximization

Our method builds on the self-supervised learning framework used by most of the recent self-supervised learning approaches [16–19, 22, 24, 25]. For a dataset with $N$ points $x_i$, each point goes through a random transformation $t^p(x_i)$ (e.g. random crop), and then forms a feature representation $z_i^p = f_\theta(t^p(x_i))$ with an encoder network $f_\theta$. We associate each image $x_i$ with its identity $y_i$, which works as a one-hot encoded label: $y_i \in \mathbb{R}^N$ and $(y_i)_d = 1$ iff $d = i$ (and zero otherwise). To match the transformations and image identities, we maximize the dependence between $z_i$ and $y_i$ such that $z_i$ is predictive of its original image. To build representations suitable for downstream tasks, we also need to penalize high-variance representations. These ideas come together in our HSIC-based objective for self-supervised learning, which we term SSL-HSIC:

$$\mathcal{L}_{\mathrm{SSL-HSIC}}(\theta) = -\mathrm{HSIC}\,(Z, Y) + \gamma\, \sqrt{\mathrm{HSIC}\,(Z, Z)}. \tag{4}$$

Unlike contrastive losses, which make the $z_i^p$ from the same $x_i$ closer and those from different $x_j$ more distant, we propose an alternative way to match different transformations of the same image with its *abstract identity* (e.g. position in the dataset).

Our objective also resembles the HSIC bottleneck for supervised learning [27] (in particular, the version of [28]), but ours uses a square root for $\mathrm{HSIC}(Z, Z)$. The square root makes the two terms on the same scale: $\mathrm{HSIC}(Z, Y)$ is effectively a dot product, and $\sqrt{\mathrm{HSIC}(Z, Z)}$ a norm, so that e.g. scaling the kernel by a constant does not change the relative amount of regularization;[2] this also gives better performance in practice.

Due to the one-hot encoded labels, we can re-write $\mathrm{HSIC}\,(Z, Y)$ as (see Appendix A)

$$\mathrm{HSIC}(Z, Y) \propto \mathbb{E}_{z_1, z_2 \sim pos}\left[ k(z_1, z_2) \right] - \mathbb{E}_{z_1} \mathbb{E}_{z_2}\left[ k(z_1, z_2) \right], \tag{5}$$

---

[1] In a slight abuse of notation, we use $\phi(x)\psi(y)^\top$ for the tensor product $\phi(x) \otimes \psi(y) \in \mathcal{F} \otimes \mathcal{G}$.

[2] Other prior work on maximizing HSIC [41, 45] used $\mathrm{HSIC}(Z, Y)/\sqrt{\mathrm{HSIC}(Z, Z)\,\mathrm{HSIC}(Y, Y)}$, or equivalently [46] the distance correlation [47]; the kernel-target alignment [48, 49] is also closely related. Here, the overall scale of either kernel does not change the objective. Our $\mathrm{HSIC}(Y, Y)$ is constant (hence absorbed in $\gamma$), and we found an additive penalty to be more stable in optimization than dividing the estimators.

where the first expectation is over the distribution of "positive" pairs (those from the same source image), and the second one is a sum over all image pairs, including their transformations. The first term in (5) pushes representations belonging to the same image identity together, while the second term keeps mean representations for each identity apart (as in Figure 2). The scaling of $\mathrm{HSIC}(Z, Y)$ depends on the choice of the kernel over $Y$, and is irrelevant to the optimization.

This form also reveals three key theoretical results. Section 3.1 shows that InfoNCE is better understood as an HSIC-based loss than a mutual information between views. Section 3.2 reveals that the dependence maximization in $\mathrm{HSIC}(Z, Y)$ can also be viewed as a form of distance metric learning, where the cluster structure is defined by the labels. Finally, $\mathrm{HSIC}(Z, Y)$ is proportional to the average kernel-based distance between the distribution of views for each source image (the maximum mean discrepancy, MMD; see Appendix B.2).

## 3.1 Connection to InfoNCE

In this section we show the connection between InfoNCE and our loss; see Appendix B for full details. We first write the latter in its infinite sample size limit (see [23] for a derivation) as

$$\mathcal{L}_{\mathrm{InfoNCE}}(\theta) = -\mathbb{E}_{z_1, z_2 \sim \mathrm{pos}}\left[k(z_1, z_2)\right] + \mathbb{E}_{z_1} \log \mathbb{E}_{z_2}\left[\exp\left(k(z_1, z_2)\right)\right], \tag{6}$$

where the last two expectations are taken over all points, and the first is over the distribution of positive pairs. The kernel $k(z_1, z_2)$ was originally formulated as a scoring function in a form of a dot product [15], and then a scaled cosine similarity [19]. Both functions are valid kernels.

Now assume that $k(z_1, z_2)$ doesn't deviate much from $\mathbb{E}_{z_2}\left[k(z_1, z_2)\right]$, Taylor-expand the exponent in (6) around $\mathbb{E}_{z_2}\left[k(z_1, z_2)\right]$, then expand $\log(1 + \mathbb{E}_{z_2}(\dots)) \approx \mathbb{E}_{z_2}(\dots)$. We obtain an $\mathrm{HSIC}(Z, Y)$-based objective:

$$\mathcal{L}_{\mathrm{InfoNCE}}(\theta) \approx \underbrace{-\mathbb{E}_{z_1, z_2 \sim \mathrm{pos}}\left[k(z_1, z_2)\right] + \mathbb{E}_{z_1}\mathbb{E}_{z_2}\left[k(z_1, z_2)\right]}_{\propto -\mathrm{HSIC}(Z, Y)} + \frac{1}{2}\underbrace{\mathbb{E}_{z_1}\left[\mathbb{V}\mathrm{ar}_{z_2}\left[k(z_1, z_2)\right]\right]}_{\text{variance penalty}}. \tag{7}$$

Since the scaling of $\mathrm{HSIC}(Z, Y)$ is irrelevant to the optimization, we assume scaling to replace $\propto$ with $=$. In the small variance regime, we can show that for the right $\gamma$,

$$-\mathrm{HSIC}\left(Z, Y\right) + \gamma \mathrm{HSIC}\left(Z, Z\right) \leq \mathcal{L}_{\mathrm{InfoNCE}}(\theta) + o(\text{variance}). \tag{8}$$

For $\mathrm{HSIC}(Z, Z) \leq 1$, we also have that

$$-\mathrm{HSIC}\left(Z, Y\right) + \gamma \mathrm{HSIC}\left(Z, Z\right) \leq \mathcal{L}_{\mathrm{SSL-HSIC}}(\theta) \tag{9}$$

due to the square root. InfoNCE and SSL-HSIC in general don't quite bound each other due to discrepancy in the variance terms, but in practice the difference is small.

Why should we prefer the HSIC interpretation of InfoNCE? Initially, InfoNCE was suggested as a variational approximation to the mutual information between two views [15]. It has been observed, however, that using tighter estimators of mutual information leads to worse performance [30]. It is also simple to construct examples where InfoNCE finds different representations while the underlying MI remains constant [30]. Alternative theories suggest that InfoNCE balances alignment of "positive" examples and uniformity of the overall feature representation [23], or that (under strong assumptions) it can identify the latent structure in a hypothesized data-generating process, akin to nonlinear ICA [50]. Our view is consistent with these theories, but doesn't put restrictive assumptions on the input data or learned representations. In Section 5 (summarized in Table 8a), we show that our interpretation gives rise to a better objective in practice.

## 3.2 Connection to metric learning

Our SSL-HSIC objective is closely related to kernel alignment [48], especially centered kernel alignment [49]. As a kernel method for distance metric learning, kernel alignment measures the agreement between a kernel function and a target function. Intuitively, the self-supervised labels $Y$ imply a cluster structure, and $\mathrm{HSIC}(Z, Y)$ estimates the degree of agreement between the learned features and this cluster structure in the kernel space. This relationship with clustering is also

established in [40, 41, 45], where labels are learned rather than features. The clustering perspective is more evident when we assume linear kernels over both $Z$ and $Y$, and $Z$ is unit length and centered:[3]

$$- \text{HSIC}(Z, Y) \propto -\frac{1}{M} Tr(Y^\top Z^\top Z Y) + Tr(Z^\top Z) - NM$$

$$= -\frac{1}{M} \sum_{i=1}^{N} \left\| \sum_{p=1}^{M} z_i^p \right\|_2^2 + \sum_{i=1}^{N} \sum_{p=1}^{M} \|z_i^p\|_2^2 - NM = \sum_{i=1}^{N} \sum_{p=1}^{M} \|z_i^p - \bar{z}_i\|_2^2 - NM, \quad (10)$$

with $M$ the number of augmentations per image and $\bar{z}_i = \sum_p z_i^p / M$ the average feature vector of the augmented views of $x_i$. We emphasize, though, that (10) assumes centered, normalized data with linear kernels; the right-hand side of (10) could be optimized by setting all $z_i^p$ to the same vector for each $i$, but this does not actually optimize $\text{HSIC}(Z, Y)$.

Equation (10) shows that we recover the spectral formulation [51] and sum-of-squares loss used in the k-means clustering algorithm from the kernel objective. Moreover, the self-supervised label imposes that the features from transformations of the same image are gathered in the same cluster. Equation (10) also allows us to connect SSL-HSIC to non-contrastive objectives such as BYOL, although the connection is subtle because of its use of predictor and target networks. If each image is augmented with two views, we can compute (10) using $\bar{z}_i \approx (z_i^1 + z_i^2)/2$, so the clustering loss becomes $\propto \sum_i \|z_i^1 - z_i^2\|_2^2$. This is exactly the BYOL objective, only that $z_i^2$ in BYOL comes from a target network. The assumption of centered and normalized features for (10) is important in the case of BYOL: without it, BYOL can find trivial solutions where all the features are collapsed to the same feature vector far away from the origin. The target network is used to prevent the collapse. SSL-HSIC, on the other hand, rules out such a solution by building the centering into the loss function, and therefore can be trained successfully without a target network or stop gradient operation.

### 3.3 Estimator of SSL-HSIC

To use SSL-HSIC, we need to correctly and efficiently estimate (4). Both points are non-trivial: the self-supervised framework implies non-i.i.d. batches (due to positive examples), while the estimator in (3) assumes i.i.d. data; moreover, the time to compute (3) is quadratic in the batch size.

First, for $\text{HSIC}(Z, Z)$ we use the biased estimator in (3). Although the i.i.d. estimator (3) results in an $O(1/B)$ bias for $B$ original images in the batch size (see Appendix A), the batch size $B$ is large in our case and therefore the bias is negligible. For $\text{HSIC}(Z, Y)$ the situation is more delicate: the i.i.d. estimator needs re-scaling, and its bias depends on the number of positive examples $M$, which is typically very small (usually 2). We propose the following estimator:

$$\widehat{\text{HSIC}}(Z, Y) = \frac{\Delta l}{N} \left( \frac{1}{BM(M-1)} \sum_{ipl} k(z_i^p, z_i^l) - \frac{1}{B^2 M^2} \sum_{ijpl} k(z_i^p, z_j^l) - \frac{1}{M-1} \right), \quad (11)$$

where $i$ and $j$ index original images, and $p$ and $l$ their random transformations; $k$ is the kernel used for latent $Z$, $l$ is the kernel used for the labels, and $\Delta l = l(i, i) - l(i, j)$ ($l$ for same labels minus $l$ for different labels). Note that due to the one-hot structure of self-supervised labels $Y$, the standard (i.i.d.-based) estimator would miss the $1/N$ scaling and the $M - 1$ correction (the latter is important in practice, as we usually have $M = 2$). See Appendix A for the derivations.

For convenience, we assume $\Delta l = N$ (any scaling of $l$ can be subsumed by $\gamma$), and optimize

$$\widehat{\mathcal{L}}_{\text{SSL-HSIC}}(\theta) = -\widehat{\text{HSIC}}(Z, Y) + \gamma \sqrt{\widehat{\text{HSIC}}(Z, Z)}. \quad (12)$$

The computational complexity of the proposed estimators is $O(B^2 M^2)$ for each mini-batch of size $B$ with $M$ augmentations. We can reduce the complexity to $O(BM)$ by using random Fourier features (RFF) [33], which approximate the kernel $k(z_1, z_2)$ with a carefully chosen random $D$-dimensional approximation $R(z_1)^\top R(z_2)$ for $R(z) : \mathbb{R}^{D_z} \to \mathbb{R}^D$, such that $k(z_1, z_2) = \mathbb{E}\left[R(z_1)^\top R(z_2)\right]$. Fourier frequencies are sampled independently for the two kernels on $Z$ in $\text{HSIC}(Z, Z)$ at each training step. We leave the details on how to construct $R(z)$ for the kernels we use to Appendix C.

---

[3]Centered $Z$ is a valid assumption for BYOL, as the target network keeps representations of views with different image identities away from each other. For high-dimensional unit vectors, this can easily lead to orthogonal representations. We also observe centered representations empirically: see Appendix B.3.

# 4 Experiments

In this section, we present our experimental setup, where we assess the performance of the representation learned with SSL-HSIC both with and without a target network. First, we train a model with a standard ResNet-50 backbone using SSL-HSIC as objective on the training set of ImageNet ILSVRC-2012 [1]. For evaluation, we retain the backbone as a feature extractor for downstream tasks. We evaluate the representation on various downstream tasks including classification, object segmentation, object detection and depth estimation.

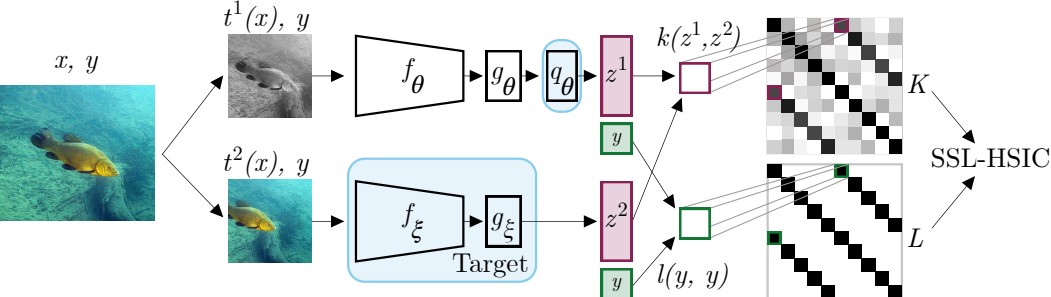

Figure 4: Architecture and SSL-HSIC objective. A self-supervised label $y$ – an indicator of the image identity – is associated with an image $x$. Image transformation functions $t$ are sampled and applied to the original image, resulting in views $t^1(x)$ and $t^2(x)$. Features $z^1$ and $z^2$ are obtained after passing the augmented views through encoder ($f$), projector ($g$), and possibly predictor ($q$) networks, while label $y$ is retained. Kernel matrices, $K$ for the latents and $L$ for the labels, are computed on the mini-batch of data; SSL-HSIC is estimated with $K$ and $L$ as in (12). The blue boxes reflect two potential options: when using a target network, $\xi$ is a moving average of $\theta$, and a predictor network $q$ is added; without the target network, $q$ is removed and $\xi$ is simply equal to $\theta$.

## 4.1 Implementation

**Architecture** Figure 4 illustrates the architecture we used for SSL-HSIC in this section. To facilitate comparison between different methods, our encoder $f_\theta$ uses the standard ResNet-50 backbone without the final classification layer. The output of the encoder is a 2048-dimension embedding vector, which is the representation used for downstream tasks. As in BYOL [25], our projector $g$ and predictor $q$ networks are 2-layer MLPs with 4096 hidden dimensions and 256 output dimensions. The outputs of the networks are batch-normalized and rescaled to unit norm before computing the loss. We use an inverse multiquadric kernel (IMQ) for the latent representation (approximated with 512 random Fourier features that are resampled at each step; see Appendix C for details) and a linear kernel for labels. $\gamma$ in (4) is set to 3. When training without a target network, unlike SimSiam [24], we do not stop gradients for either branch. If the target network is used, its weights are an exponential moving average of the online network weights. We employ the same schedule as BYOL [25], $\tau = 1 - 0.01 \cdot (\cos(\pi t/T) + 1)/2$ with $t$ the current step and $T$ the total training steps.

**Image augmentation** Our method uses the same data augmentation scheme as BYOL (see Appendix D.1). Briefly, we first draw a random patch from the original image and resize it to $224 \times 224$. Then, we apply a random horizontal flip, followed by color jittering, consisting of a random sequence of brightness, contrast, saturation, hue adjustments, and an optional grayscale conversion. Finally Gaussian blur and solarization are applied, and the view is normalized with ImageNet statistics.

**Optimization** We train the model with a batch size of 4096 on 128 Cloud TPU v4 cores. Again, following [19, 25], we use the LARS optimizer [52] with a cosine decay learning rate schedule over 1000 epochs. The base learning rate to all of our experiments is $0.4$ and it is scaled linearly [53] with the batch size $lr = 0.4 \times batch\_size/256$. All experiments use weight decay of $10^{-6}$.

**Learning kernel parameters** We use a linear kernel for labels, since the type of kernel only scales (12). Our inverse multiquadric kernel for the latent $Z$ has an additional kernel scale parameter. We optimize this along with all other parameters, but regularize it to maximize the entropy of the distribution $k_\sigma(s)$, where $s_{ij} = \|z_i - z_j\|^2$; this amounts to maximizing $\log\|k'_\sigma(s)\|^2$ (Appendix D.1.2).

## 4.2 Evaluation Results

**Linear evaluation on ImageNet** Learned features are evaluated with the standard linear evaluation protocol commonly used in evaluating self-supervised learning methods [15–22, 25]. Table 1 reports the top-1 and top-5 accuracies obtained with SSL-HSIC on ImageNet validation set, and compares to previous self-supervised learning methods. Without a target network, our method reaches 72.2% top-1 and 90.7% top-5 accuracies. Unlike BYOL, the SSL-HSIC objective prevents the network from finding trivial solutions as explained in Section 3.2. Adding the target network, our method outperforms most previous methods, achieving top-1 accuracy of 74.8% and top-5 accuracy of 92.2%. The fact that we see performance gains from adopting a target network suggests that its effect is not yet well understood, although note discussion in [25] which points to its stabilizing effect.

Table 1: Linear evaluation on the ImageNet validation set.

|  | Top-1(%) | Top-5(%) |
|---|---|---|
| Supervised [54] | 75.9 | 92.8 |
| SimCLR [19] | 69.3 | 89.0 |
| MoCo v2 [20] | 71.1 | 90.1 |
| BYOL [25] | 74.3 | 91.6 |
| SwAV [18] | **75.3** | - |
| Barlow Twins [22] | 73.2 | 91.0 |
| SSL-HSIC (w/o target) | 72.2 | 90.7 |
| SSL-HSIC (w/ target) | 74.8 | **92.2** |

Table 2: Fine-tuning on 1%, 10% and 100% of the ImageNet training set and evaluating on the validation set.

|  | Top-1(%) | | | Top-5(%) | | |
|---|---|---|---|---|---|---|
|  | 1% | 10% | 100% | 1% | 10% | 100% |
| Supervised [54] | 25.4 | 56.4 | 75.9 | 48.4 | 80.4 | 92.8 |
| SimCLR [19] | 48.3 | 65.6 | 76.0 | 75.5 | 87.8 | 93.1 |
| BYOL [25] | 53.2 | 68.8 | **77.7** | 78.4 | 89.0 | **93.9** |
| SwAV [18] | 53.9 | 70.2 | - | 78.5 | **89.9** | - |
| Barlow Twins [22] | **55.0** | 69.7 | - | **79.2** | 89.3 | - |
| SSL-HSIC (w/o target) | 45.3 | 65.5 | 76.4 | 72.7 | 87.5 | 93.2 |
| SSL-HSIC (w/ target) | 52.1 | 67.9 | 77.2 | 77.7 | 88.6 | 93.6 |

**Semi-supervised learning on ImageNet** We fine-tune the network pretrained with SSL-HSIC on 1%, 10% and 100% of ImageNet, using the same ImageNet splits as SimCLR [19]. Table 2 summarizes the semi-supervised learning performance. Our method, with or without a target network, has competitive performance in both data regimes. The target network has the most impact on the small-data regime, with 1% labels.

Table 3: Comparison of transfer learning performance on 12 image datasets. Supervised-IN is trained on ImageNet with supervised pretrainining. Random init trains on individual dataset with randomly initialized weights. MPCA refers to mean per-class accuracy; AP50 is average precision at IoU=0.5.

| Dataset
Metric | Birdsnap
Top-1 | Caltech101
MPCA | Cifar10
Top-1 | Cifar100
Top-1 | DTD
Top-1 | Aircraft
MPCA | Food
Top-1 | Flowers
MPCA | Pets
MPCA | Cars
Top-1 | SUN397
Top-1 | VOC2007
AP50 |
|---|---|---|---|---|---|---|---|---|---|---|---|---|
| *Linear:* | | | | | | | | | | | | |
| Supervised-IN [19] | 53.7 | **94.5** | **93.6** | 78.3 | 74.9 | **61.0** | 72.3 | 94.7 | **91.5** | **67.8** | **61.9** | 82.8 |
| SimCLR [19] | 37.4 | 90.3 | 90.6 | 71.6 | 74.5 | 50.3 | 68.4 | 90.3 | 83.6 | 50.3 | 58.8 | 80.5 |
| BYOL [25] | 57.2 | 94.2 | 91.3 | **78.4** | 75.5 | 60.6 | 75.3 | **96.1** | 90.4 | 66.7 | 62.2 | 82.5 |
| SSL-HSIC (w/o target) | 50.6 | 92.3 | 91.5 | 75.9 | 75.3 | 57.9 | 73.6 | 95.0 | 88.2 | 59.3 | 61.0 | 81.4 |
| SSL-HSIC (w/ target) | **57.8** | 93.5 | 92.3 | 77.0 | **76.2** | 58.5 | **75.6** | 95.4 | 91.2 | 62.6 | 61.8 | **83.3** |
| *Fine-tuned:* | | | | | | | | | | | | |
| Supervised-IN [19] | 75.8 | 93.3 | 97.5 | **86.4** | 74.6 | 86.0 | 88.3 | **97.6** | 92.1 | 92.1 | **94.3** | 85.0 |
| Random init [19] | **76.1** | 72.6 | 95.9 | 80.2 | 64.8 | 85.9 | 86.9 | 92.0 | 81.5 | 91.4 | 53.6 | 67.3 |
| SimCLR [19] | 75.9 | 92.1 | 97.7 | 85.9 | 73.2 | 88.1 | 88.2 | 97.0 | 89.2 | 91.3 | 63.5 | 84.1 |
| BYOL [25] | 76.3 | **93.8** | **97.8** | 86.1 | **76.2** | 88.1 | **88.5** | 97.0 | 91.7 | 91.6 | 63.7 | **85.4** |
| SSL-HSIC (w/o target) | 73.1 | 91.5 | 97.4 | 85.3 | 75.3 | 87.1 | 87.5 | 96.4 | 90.6 | 91.6 | 62.2 | 84.1 |
| SSL-HSIC (w/ target) | 74.9 | **93.8** | **97.8** | 84.7 | 75.4 | **88.9** | 87.7 | 97.3 | 91.7 | 91.8 | 61.7 | 84.1 |

**Transfer to other classification tasks** To investigate the generality of the representation learned with SSL-HSIC, we evaluate the transfer performance for classification on 12 natural image datasets [55–64] using the same procedure as [19, 25, 65]. Table 3 shows the top-1 accuracy of the linear evaluation and fine-tuning performance on the test set. SSL-HSIC gets state-of-the-art performance on 3 of the classification tasks and reaches strong performance on others for this benchmark, indicating the learned representations are robust for transfer learning.

**Transfer to other vision tasks** To test the ability of transferring to tasks other than classification, we fine-tune the network on semantic segmentation, depth estimation and object detection tasks. We use Pascal VOC2012 dataset [58] for semantic segmentation, NYU v2 dataset [66] for depth estimation and COCO [67] for object detection. Object detection outputs either bounding box or object segmentation (instance segmentation). Details of the evaluations setup is in Appendix D.2. Table 4 and Table 5 shows that SSL-HSIC achieves competitive performance on all three vision tasks.

Table 4: Fine-tuning performance on semantic segmentation and depth estimation. Mean Intersection over Union (mIoU) is reported for semantic segmentation. Relative error (rel), root mean squared error (rms), and the percent of pixels (pct) where the error is below $1.25^n$ thresholds are reported for depth estimation.

| | VOC2012 | NYU v2 | | | | |
|---|---|---|---|---|---|---|
| Method | mIoU | pct.$< 1.25$ | pct.$< 1.25^2$ | pct.$< 1.25^3$ | rms | rel |
| Supervised-IN | 74.4 | 81.1 | 95.3 | 98.8 | 0.573 | **0.127** |
| SimCLR | 75.2 | 83.3 | 96.5 | 99.1 | 0.557 | 0.134 |
| BYOL | **76.3** | **84.6** | 96.7 | 99.1 | 0.541 | 0.129 |
| SSL-HSIC(w/o target) | 74.9 | 84.1 | 96.7 | **99.2** | **0.539** | 0.130 |
| SSL-HSIC(w/ target) | 76.0 | 83.8 | **96.8** | 99.1 | 0.548 | 0.130 |

Table 5: Fine-tuning performance on COCO object detection tasks. Precision, averaged over 10 IoU (Intersection over Union) thresholds, is reported for both bounding box and object segmentation.

| Method | $AP^{bb}$ | $AP^{mk}$ |
|---|---|---|
| Supervised | 39.6 | 35.6 |
| SimCLR | 39.7 | 35.8 |
| MoCo v2 | 40.1 | 36.3 |
| BYOL | **41.6** | 37.2 |
| SwAV | **41.6** | **37.8** |
| SSL-HSIC(w/o target) | 40.5 | 36.3 |
| SSL-HSIC(w/ target) | 41.3 | 36.8 |

# 5 Ablation Studies

We present ablation studies to gain more intuition on SSL-HSIC. Here, we use a ResNet-50 backbone trained for 100 epochs on ImageNet, and evaluate with the linear protocol unless specified.

**ResNet architectures** In this ablation, we investigate the performance of SSL-HSIC with wider and deeper ResNet architecture. Figure 1 and Table 6 show our main results. The performance of SSL-HSIC gets better with larger networks. We used the supervised baseline from [25] which our training framework is based on ([19] reports lower performance). The performance gap between SSL-HSIC and the supervised baseline diminishes with larger architectures. In addition, Table 7 presents the semi-supervise learning results with subsets 1%, 10% and 100% of the ImageNet data.

Table 6: Top-1 and top-5 accuracies for different ResNet architectures using linear evaluation protocol.

| | SSL-HSIC | | BYOL[25] | | Sup.[25] | |
|---|---|---|---|---|---|---|
| ResNet | Top1 | Top5 | Top1 | Top5 | Top1 | Top5 |
| 50 (1x) | 74.8 | 92.2 | 74.3 | 91.6 | 76.4 | 92.9 |
| 50 (2x) | 77.9 | 94.0 | 77.4 | 93.6 | 79.9 | 95.0 |
| 50 (4x) | 79.1 | 94.5 | 78.6 | 94.2 | 80.7 | 95.3 |
| 200 (2x) | 79.6 | 94.8 | 79.6 | 94.9 | 80.1 | 95.2 |

Table 7: Top-1 and top-5 accuracies for different ResNet architectures using semi-supervised fine-tuning.

| | Top1 | | | Top5 | | |
|---|---|---|---|---|---|---|
| ResNet | 1% | 10% | 100% | 1% | 10% | 100% |
| 50 (1x) | 52.1 | 67.9 | 77.2 | 77.7 | 88.6 | 93.6 |
| 50 (2x) | 61.2 | 72.6 | 79.3 | 83.8 | 91.2 | 94.7 |
| 50 (4x) | 67.0 | 75.4 | 79.7 | 87.4 | 92.5 | 94.8 |
| 200(2x) | 69.0 | 76.3 | 80.5 | 88.3 | 92.9 | 95.2 |

**Regularization term** We compare performance of InfoNCE with SSL-HSIC in Table 8a since they can be seen as approximating the same $HSIC(Z, Y)$ objective but with different forms of regularization. We reproduce the InfoNCE result in our codebase, using the same architecture and data augmentiation as for SSL-HSIC. Trained for 100 epochs (without a target network), InfoNCE achieves 66.0% top-1 and 86.9% top-5 accuracies, which is better than the result reported in [19]. For comparison, SSL-HSIC reaches 66.7% top-1 and 87.6% top-5 accuracies. This suggests that the regularization employed by SSL-HSIC is more effective.

**Kernel type** We investigate the effect of using different a kernel on latents $Z$. Training without a target network or random Fourier feature approximation, the top-1 accuracies for linear, Gaussian, and inverse multiquadric (IMQ) kernels are $65.27\%$, $66.67\%$ and $66.72\%$ respectively. Non-linear kernels indeed improve the performance; Gaussian and IMQ kernels reach very similar performance for 100 epochs. We choose IMQ kernel for longer runs, because its heavy-tail property can capture more signal when points are far apart.

**Number of RFF Features** Table 8b shows the performance of SSL-HSIC with different numbers of Fourier features. The RFF approximation has a minor impact on the overall performance, as long as we resample them; fixed sets of features performed poorly. Our main result picked 512 features, for substantial computational savings with minor loss in accuracy.

**Batch size** Similar to most of the self-supervised learning methods [19, 25], SSL-HSIC benefits from using a larger batch size during training. However, the drop of performance from using smaller batch size is not as pronounced as it is in SimCLR[19] as shown in Table 8c.

Table 8: Linear evaluation results when varying different hyperparameters.

(a) Regularization

| | Top-1 | Top-5 |
|---|---|---|
| SSL-HSIC | 66.7 | 87.6 |
| InfoNCE | 66.0 | 86.9 |

(b) # Fourier features

| # RFFs | Top-1(%) |
|---|---|
| 64 | 66.0 |
| 128 | 66.2 |
| 256 | 66.2 |
| 512 | 66.4 |
| 1024 | 66.5 |
| 2048 | 66.5 |
| No Approx. | 66.7 |

(c) Batch size

| | Top-1(%) | |
|---|---|---|
| Batch Size | SSL-HSIC | SimCLR |
| 256 | 63.7 | 57.5 |
| 512 | 65.6 | 60.7 |
| 1024 | 66.7 | 62.8 |
| 2048 | 67.1 | 64.0 |
| 4096 | 66.7 | 64.6 |

(d) Projector/predictor size

| Output Dim | Top-1(%) |
|---|---|
| 64 | 65.4 |
| 128 | 66.0 |
| 256 | 66.4 |
| 512 | 66.6 |
| 1024 | 66.6 |

**Projector and predictor output size** Table 8d shows the performance when using different output dimension for the projector/predictor networks. The performance saturates at 512 dimensions.

# 6 Conclusions

We introduced SSL-HSIC, a loss function for self-supervised representation learning based on kernel dependence maximization. We provided a unified view on various self-supervised learning losses: we proved that InfoNCE, a lower bound of mutual information, actually approximates SSL-HSIC with a variance-based regularization, and we can also interpret SSL-HSIC as metric learning where the cluster structure is imposed by the self-supervised label, of which the BYOL objective is a special case. We showed that training with SSL-HSIC achieves performance on par with the state-of-the-art on the standard self-supervised benchmarks.

Although using the image identity as self-supervised label provides a good inductive bias, it might not be wholly satisfactory; we expect that some images pairs are in fact more similar than others, based e.g. on their ImageNet class label. It will be interesting to explore methods that combine label structure discovery with representation learning (as in SwAV [18]). In this paper, we only explored learning image representations, but in future work SSL-HSIC can be extended to learning structure for $Y$ as well, building on existing work [41, 45].

# Broader impact

Our work concentrates on providing a more theoretically grounded and interpretable loss function for self-supervised learning. A better understanding of self-supervised learning, especially through more interpretable learning dynamics, is likely to lead to better and more explicit control over societal biases of these algorithms. SSL-HSIC yields an alternative, clearer understanding of existing self-supervised methods. As such, it is unlikely that our method introduces further biases than those already present for self-supervised learning.

The broader impacts of the self-supervised learning framework is an area that has not been studied by the AI ethics community, but we think it calls for closer inspection. An important concern for fairness of ML algorithms is dataset bias. ImageNet is known for a number of problems such as offensive annotations, non-visual concepts and lack of diversity, in particular for underrepresented groups. Existing works and remedies typically focus on label bias. Since SSL doesn't use labels, however, the type and degree of bias could be very different from that of supervised learning. To mitigate the risk of dataset bias, one could employ dataset re-balancing to correct sampling bias [68] or completely exclude human images from the dataset while achieving the same performance [69].

A new topic to investigate for self-supervised learning is how the bias/unbiased representation could be transferred to downstream tasks. We are not aware of any work in this direction. Another area of concern is security and robustness. Compared to supervised learning, self-supervised learning typically involves more intensive data augmentation such as color jittering, brightness adjustment, etc. There is some initial evidence suggesting self-supervised learning improves model robustness [70]. However, since data augmentation can either be beneficial [71] or detrimental [72] depending on the type of adversarial attacks, more studies are needed to assess its role for self-supervised learning.

## Acknowledgments and Disclosure of Funding

The authors would like to thank Olivier J. Hénaff for valuable feedback on the manuscript and the help for evaluating object detection task. We thank Aaron Van den Oord and Oriol Vinyals for providing valuable feedback on the manuscript. We are grateful to Yonglong Tian, Ting Chen and the BYOL authors for the help with reproducing baselines and evaluating downstream tasks.

This work was supported by DeepMind, the Gatsby Charitable Foundation, the Wellcome Trust, NSERC, and the Canada CIFAR AI Chairs program.

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
