# Appendices

## A  HSIC estimation in the self-supervised setting

Estimators of HSIC typically assume i.i.d. data, which is not the case for self-supervised learning – the positive examples are not independent. Here we show how to adapt our estimators to the self-supervision setting.

### A.1  Exact form of HSIC(Z, Y)

Starting with $\text{HSIC}(Z, Y)$, we assume that the "label" $y$ is a one-hot encoding of the data point, and all $N$ data points are sampled with the same probability $1/N$. With a one-hot encoding, any kernel that is a function of $y_i^\top y_j$ or $\|y_i - y_j\|$ (e.g. linear, Gaussian or IMQ) have the form

$$l(y_i, y_j) = \begin{cases} l_1 & y_i = y_j, \\ l_0 & \text{otherwise} \end{cases} \equiv \Delta l\, \mathbb{I}(y_i = y_j) + l_0 \tag{13}$$

for some $\Delta l = l_1 - l_0$.

**Theorem A.1.** *For a dataset with $N$ original images sampled with probability $1/N$, and a kernel over image identities defined as in* (13), $\text{HSIC}(Z, Y)$ *takes the form*

$$\text{HSIC}(Z, Y) = \frac{\Delta l}{N} \mathbb{E}_{Z, Z' \sim \text{pos}} [k(Z, Z')] - \frac{\Delta l}{N} \mathbb{E}\, [k(Z, Z')]\ , \tag{14}$$

*where $Z, Z' \sim \text{pos}$ means $p_{\text{pos}}(Z, Z') = \sum_i p(i) p(Z|i) p(Z'|i)$ for image probability $p(i) = 1/N$.*

*Proof.* We compute HSIC (defined in (2)) term by term. Starting from the first, and denoting independent copies of $Z, Y$ with $Z', Y'$,

$$
\begin{aligned}
\mathbb{E}\, [k(Z, Z') l(Y, Y')] &= \Delta l\, \mathbb{E}\, [k(Z, Z') \mathbb{I}[Y = Y']] + l_0\, \mathbb{E}\, [k(Z, Z')] \\
&= \Delta l \sum_{i=1}^{N} \sum_{j=1}^{N} \mathbb{E}_{Z|y_i, Z'|y_j} \left[ \frac{1}{N^2} k(Z, Z') \mathbb{I}[y_i = y_j] \right] + l_0\, \mathbb{E}\, [k(Z, Z')] \\
&= \frac{\Delta l}{N} \sum_{i=1}^{N} \mathbb{E}_{Z|y_i, Z'|y_i} \left[ \frac{1}{N} k(Z, Z') \right] + l_0\, \mathbb{E}\, [k(Z, Z')] \\
&= \frac{\Delta l}{N} \mathbb{E}_{Z, Z' \sim \text{pos}} [k(Z, Z')] + l_0\, \mathbb{E}\, [k(Z, Z')]\ ,
\end{aligned}
$$

where $\mathbb{E}_{Z, Z' \sim \text{pos}}$ is the expectation over positive examples (with $Z$ and $Z'$ are sampled independently conditioned on the "label").

The second term, due to the independence between $Z'$ and $Y''$, becomes

$$\mathbb{E}\, [k(Z, Z') l(Y, Y'')] = \mathbb{E}_{ZY} \mathbb{E}_{Z'} \left[ k(Z, Z') \left( \frac{\Delta l}{N} + l_0 \right) \right] = \left( \frac{\Delta l}{N} + l_0 \right) \mathbb{E}\, [k(Z, Z')]\ .$$

And the last term becomes identical to the second one,

$$\mathbb{E}\, [k(Z, Z')]\, \mathbb{E}\, [l(Y, Y')] = \left( \frac{\Delta l}{N} + l_0 \right) \mathbb{E}\, [k(Z, Z')]\ .$$

Therefore, we can write $\text{HSIC}(Z, Y)$ as

$$
\begin{aligned}
\text{HSIC}(Z, Y) &= \mathbb{E}\, [k(Z, Z') l(Y, Y')] - 2\, \mathbb{E}\, [k(Z, Z') l(Y, Y'')] + \mathbb{E}\, [k(Z, Z')]\, \mathbb{E}\, [l(Y, Y')] \\
&= \frac{\Delta l}{N} \mathbb{E}_{Z, Z' \sim \text{pos}} [k(Z, Z')] - \frac{\Delta l}{N} \mathbb{E}\, [k(Z, Z')]\ ,
\end{aligned}
$$

as the terms proportional to $l_0$ cancel each other out.  $\square$

The final form of $\text{HSIC}(Z, Y)$ shows that the $Y$ kernel and dataset size come in only as pre-factors. To make the term independent of the dataset size (as long as it is finite), we can assume $\Delta l = N$, such that

$$\text{HSIC}(Z, Y) = \mathbb{E}_{Z, Z' \sim \text{pos}} [k(Z, Z')] - \mathbb{E}\, [k(Z, Z')]\ .$$

## A.2 Estimator of HSIC(Z, Y)

**Theorem A.2.** *In the assumptions of Theorem A.1, additionally scale the $Y$ kernel to have $\Delta l = N$, and the $Z$ kernel to be $k(z,z) = 1$. Assume that the batch is sampled as follows: $B < N$ original images are sampled without replacement, and for each image $M$ positive examples are sampled independently (i.e., the standard sampling scheme in self-supervised learning). Then denoting each data point $z_i^p$ for "label" $i$ and positive example $p$,*

$$\widehat{\text{HSIC}}(Z,Y) = \left( \frac{M}{M-1} + \frac{N-1}{N(B-1)} - \frac{M}{N(M-1)} \right) \frac{1}{BM^2} \sum_{ipl} k(z_i^p, z_i^l) \tag{15}$$

$$- \frac{B(N-1)}{(B-1)N} \frac{1}{B^2 M^2} \sum_{ijpl} k(z_i^p, z_j^l) - \frac{N-1}{N(M-1)}. \tag{16}$$

*is an unbiased estimator of* (14).

While we assumed that $k(z,z) = 1$ for simplicity, any change in the scaling would only affect the constant term (which is irrelevant for gradient-based learning). Recalling that $|k(z,z')| \le \max(k(z,z), k(z',z'))$, we can then obtain a slightly biased estimator from Theorem A.2 by simply discarding small terms:

**Corollary A.2.1.** *If $|k(z,z')| \le 1$ for any $z, z'$, then*

$$\widehat{\text{HSIC}}(Z,Y) = \frac{1}{BM(M-1)} \sum_{ipl} k(z_i^p, z_i^l) - \frac{1}{B^2 M^2} \sum_{ijpl} k(z_i^p, z_j^l) - \frac{1}{M-1} \tag{17}$$

*has a $O(1/B)$ bias.*

*Proof of Theorem A.2.* To derive an unbiased estimator, we first compute expectations of two sums: one over all positives examples (same $i$) and one over all data points.

Starting with the first,

$$\mathbb{E}\left[ \frac{1}{BM^2} \sum_{ipl} k(z_i^p, z_i^l) \right] = \mathbb{E}\left[ \frac{1}{BM^2} \sum_{ip, l \neq p} k(z_i^p, z_i^l) \right] + \mathbb{E}\left[ \frac{1}{BM^2} \sum_{ip} k(z_i^p, z_i^p) \right] \tag{18}$$

$$= \frac{M-1}{M} \mathbb{E}_{Z, Z' \sim \text{pos}} \left[ k(Z, Z') \right] + \frac{1}{M}. \tag{19}$$

As for the second sum,

$$\mathbb{E}\left[ \frac{1}{B^2 M^2} \sum_{ijpl} k(z_i^p, z_j^l) \right] = \mathbb{E}\left[ \frac{1}{B^2 M^2} \sum_{i, j \neq i, pl} k(z_i^p, z_j^l) \right] + \mathbb{E}\left[ \frac{1}{B^2 M^2} \sum_{ipl} k(z_i^p, z_i^l) \right].$$

The first term is tricky: $\mathbb{E}\left[ k(z_i^p, z_j^l) \right] \neq \mathbb{E}\left[ k(Z, Z') \right]$ because we sample without replacement. But we know that $p(y, y') = p(y)p(y'|y) = 1/(N(N-1))$, therefore for $i \neq j$

$$\mathbb{E}\, k(z_i^p, z_j^l) = \sum_{y, y' \neq y} \frac{1}{N(N-1)} \mathbb{E}_{Z|y, Z'|y'}\, k(Z, Z') \tag{20}$$

$$= \sum_{yy'} \frac{1}{N(N-1)} \mathbb{E}_{Z|y, Z'|y'}\, k(Z, Z') - \sum_{y} \frac{1}{N(N-1)} \mathbb{E}_{Z|y, Z'|y} k(Z, Z') \tag{21}$$

$$= \frac{N}{N-1} \mathbb{E}\, k(Z, Z') - \frac{1}{N-1} \mathbb{E}_{Z, Z' \sim \text{pos}} k(Z, Z'). \tag{22}$$

Using the expectations for $ipl$ and $ijpl$,

$$\mathbb{E}\frac{1}{B^2M^2}\sum_{ijpl}k(z_i^p,z_j^l)=\mathbb{E}\frac{1}{B^2M^2}\sum_{i,j\neq i,pl}k(z_i^p,z_j^l)+\mathbb{E}\frac{1}{B^2M^2}\sum_{ipl}k(z_i^p,z_j^l) \tag{23}$$

$$=\frac{B-1}{B(N-1)}\left(N\,\mathbb{E}\,k(Z,Z')-\mathbb{E}_{Z,Z'\sim\mathrm{pos}}k(Z,Z')\right)+\frac{M-1}{BM}\mathbb{E}_{Z,Z'\sim\mathrm{pos}}k(Z,Z')+\frac{1}{BM} \tag{24}$$

$$=\frac{(B-1)N}{B(N-1)}\mathbb{E}\,k(Z,Z')-\frac{B-1}{B(N-1)}\mathbb{E}_{Z,Z'\sim\mathrm{pos}}k(Z,Z')+\frac{M-1}{BM}\mathbb{E}_{Z,Z'\sim\mathrm{pos}}k(Z,Z')+\frac{1}{BM} \tag{25}$$

$$=\frac{(B-1)N}{B(N-1)}\mathbb{E}\,k(Z,Z')+\frac{1}{B}\left(\frac{M-1}{M}-\frac{B-1}{N-1}\right)\mathbb{E}_{Z,Z'\sim\mathrm{pos}}k(Z,Z')+\frac{1}{BM}\,. \tag{26}$$

Combining (18) and (23) shows that (15) is indeed an unbiased estimator. $\qquad\square$

It's worth noting that the i.i.d. estimator (3) is flawed for $\mathrm{HSIC}(Z,Y)$ for two reasons: first, it misses the $1/N$ scaling of $\mathrm{HSIC}(Z,Y)$ (however, it's easy to fix by rescaling); second, it misses the $1/(M(M-1))$ correction for the $ipl$ sum. As we typically have $M=2$, the latter would result in a large bias for the (scaled) i.i.d. estimator.

### A.3 Estimator of HSIC(Z, Z)

Before discussing estimators of $\mathrm{HSIC}(Z,Z)$, note that it takes the following form:

$$\mathrm{HSIC}(Z,Z)=\mathbb{E}\left[k(Z,Z')^2\right]-2\mathbb{E}_Z\left[\mathbb{E}_{Z'}\left[k(Z,Z')\right]^2\right]+\left(\mathbb{E}\left[k(Z,Z')\right]\right)^2\,.$$

This is because $X$ and $Y$ in $\mathrm{HSIC}(X,Y)$ become the *same* random variable, so $p(X,Y)=p_Z(X)\delta(X-Y)$ (see [28], Appendix A).

**Theorem A.3.** *Assuming $k(z,z')\leq 1$ for any $z,z'$, the i.i.d. HSIC estimator by Gretton et al. [26],*

$$\widehat{\mathrm{HSIC}}(Z,Z)=\frac{1}{(BM-1)^2}\mathrm{Tr}(KHKH)\,,$$

*where $H=I-\frac{1}{BM}\mathbf{1}\mathbf{1}^\top$, has a $O(1/B)$ bias for the self-supervised sampling scheme.*

*Proof.* First, observe that

$$\mathrm{Tr}(KHKH)=\mathrm{Tr}(KK)-\frac{2}{BM}\mathbf{1}^\top KK\mathbf{1}+\frac{1}{B^2M^2}\left(\mathbf{1}^\top K\mathbf{1}\right)^2\,.$$

Starting with the first term, and using again the result of (20) for sampling without replacement,

$$\mathbb{E}\left[\mathrm{Tr}(KK)\right]=\mathbb{E}\left[\sum_{ijpl}k(z_i^p,z_j^l)^2\right]=\mathbb{E}\left[\sum_{i,j\neq i,pl}k(z_i^p,z_j^l)^2\right]+\mathbb{E}\left[\sum_{ipl}k(z_i^p,z_j^l)^2\right]$$

$$=\frac{B(B-1)M^2}{N-1}\left(N\,\mathbb{E}\,k(Z,Z')^2-\mathbb{E}_{Z,Z'\sim\mathrm{pos}}k(Z,Z')^2\right)+\mathbb{E}\sum_{ipl}k(z_i^p,z_j^l)^2$$

$$=B^2M^2\,\mathbb{E}\,k(Z,Z')^2+O(BM^2)\,.$$

Similarly, the expectation of the second term is

$$\mathbb{E}\,\mathbf{1}^\top KK\mathbf{1}=\mathbb{E}\sum_{ijqpld}k(z_i^p,z_q^d)k(z_j^l,z_q^d)$$

$$=\mathbb{E}\sum_{i,j\neq i,q\neq\{i,j\},pld}k(z_i^p,z_q^d)k(z_j^l,z_q^d)+O(B^2M^3)\,.$$

Here we again need to take sampling without replacement into account, and again it will produce a very small correcton term. For $i \neq j \neq q$, repeating the calculation in (20),

$$\mathbb{E}\, k(z_i^p, z_j^l)k(z_i^p, z_q^d) = \sum_{y, y' \neq y, y'' \neq \{y, y'\}} \frac{1}{N(N-1)(N-2)} \mathbb{E}_{Z|y, Z'|y', Z''|Y''}\, k(Z, Z')k(Z, Z'')$$
$$= \mathbb{E}\, k(Z, Z')k(Z, Z'') + O(1/N)\,.$$

As $B < N$, we obtain that

$$\mathbb{E}\, \mathbf{1}^\top KK\mathbf{1} = B(B-1)(B-2)M^3\, \mathbb{E}\, k(Z, Z')k(Z, Z'') + O(B^2 M^3)\,.$$

Finally, repeating the same argument for sampling without replacement,

$$\mathbb{E}\left(\mathbf{1}^\top K\mathbf{1}\right)^2 = \mathbb{E} \sum_{ijqrpldf} k(z_i^p, z_j^l)k(z_q^d, z_r^f)$$
$$= \mathbb{E} \sum_{i, j \neq i, q \neq \{i, j\}, r \neq \{i, j, q\}, pldf} k(z_i^p, z_j^l)k(z_q^d, z_r^f) + O(B^3 M^4)$$
$$= B(B-1)(B-2)(B-3)M^4\, \mathbb{E}\, k(Z, Z')k(Z'', Z''') + O(B^3 M^4)\,.$$

Combining all terms together, and expressing $B(B-1)$ (and similar) terms in big-O notation,

$$\mathbb{E}\frac{\mathrm{Tr}(KHKH)}{(BM-1)^2} = \mathbb{E}\left(k(Z, Z')^2 - 2\, k(Z, Z')k(Z, Z'') + k(Z, Z')k(Z'', Z''')\right) + O\left(\frac{1}{B}\right)$$
$$= \mathbb{E}\, k(Z, Z')^2 - 2\mathbb{E}_Z\left(\mathbb{E}_{Z'}\, k(Z, Z')\right)^2 + \left(\mathbb{E}\, k(Z, Z')\right)^2 + O\left(\frac{1}{B}\right)$$
$$= \mathrm{HSIC}(Z, Z) + O\left(\frac{1}{B}\right)\,.$$

$\square$

Essentially, having $M$ positive examples for the batch size of $BM$ changes the bias from $O(1/(BM))$ (i.i.d. case) to $O(1/B)$. Finally, note that even if $\widehat{\mathrm{HSIC}}(Z, Z)$ is unbiased, its square root is not.

# B  Theoretical properties of SSL-HSIC

## B.1  InfoNCE connection

To establish the connection with InfoNCE, define it in terms of expectations:

$$\mathcal{L}_{\mathrm{InfoNCE}}(\theta) = \mathbb{E}_Z\left[\log \mathbb{E}_{Z'}\left[\exp\left(k(Z, Z')\right)\right]\right] - \mathbb{E}_{Z, Z' \sim \mathrm{pos}}\left[k(Z, Z')\right]\,. \tag{27}$$

To clarify the reasoning in the main text, we can Taylor expand the exponent in (27) around $\mu_1 \equiv \mathbb{E}_{Z'}\left[k(Z, Z')\right]$. For $k(Z, Z') \approx \mu_1$,

$$\mathbb{E}_Z\left[\log \mathbb{E}_{Z'}\left[\exp\left(k(Z, Z')\right)\right]\right]$$
$$\approx \mathbb{E}_Z\left[\mu_1\right] + \mathbb{E}_Z\left[\log \mathbb{E}_{Z'}\left[1 + k(Z, Z') - \mu_1 + \frac{(k(Z, Z') - \mu_1)^2}{2}\right]\right]$$
$$= \mathbb{E}_Z\left[\mu_1\right] + \mathbb{E}_Z\left[\log \mathbb{E}_{Z'}\left[1 + \frac{(k(Z, Z') - \mu_1)^2}{2}\right]\right]\,.$$

Now expanding $\log(1 + x)$ around zero,

$$\mathbb{E}_Z\left[\log \mathbb{E}_{Z'}\left[\exp\left(k(Z, Z')\right)\right]\right] \approx \mathbb{E}_Z\left[\mu_1\right] + \mathbb{E}_Z\mathbb{E}_{Z'}\left[\frac{(k(Z, Z') - \mu_1)^2}{2}\right]$$
$$= \mathbb{E}_Z\mathbb{E}_{Z'}\left[k(Z, Z')\right] + \frac{1}{2}\mathbb{E}_Z\left[\mathbb{V}\mathrm{ar}_{Z'}\left[k(Z, Z')\right]\right]\,.$$

The approximate equality relates to expectations over higher order moments, which are dropped. The expression gives the required intuition behind the loss, however: when the variance in $k(Z, Z')$ w.r.t. $Z'$ is small, InfoNCE combines $-\text{HSIC}(Z, Y)$ and a variance-based penalty. In general, we can always write (assuming $\Delta l = N$ in $\text{HSIC}(Z, Y)$ as before)

$$\mathcal{L}_{\text{InfoNCE}}(\theta) = -\text{HSIC}(Z, Y) + \mathbb{E}_Z \left[ \log \mathbb{E}_{Z'} \left[ \exp \left( k(Z, Z') - \mu_1 \right) \right] \right] . \tag{28}$$

In the small variance regime, InfoNCE also bounds an HSIC-based loss. To show this, we will need a bound on $\exp(x)$:

**Lemma B.0.1.** *For $0 < \alpha \leq 1/4$ and $x \geq - \left( 1 + \sqrt{1 - 4\alpha} \right) /(2\alpha)$,*

$$\exp(x) \geq 1 + x + \alpha \, x^2 . \tag{29}$$

*Proof.* The quadratic equation $1 + x + \alpha \, x^2$ has two roots ($x_1 \leq x_2$):

$$x_{1,2} = \frac{\pm \sqrt{1 - 4\alpha} - 1}{2\alpha} .$$

Both roots are real, as $\alpha \leq 1/4$. Between $x_1$ and $x_2$, (29) holds trivially as the rhs is negative.

For $x \geq -2$ and $\alpha \leq 1/4$,

$$\exp(x) \geq 1 + x + \frac{x^2}{2!} + \frac{x^3}{3!} + \frac{x^4}{4!} + \frac{x^5}{5!} \geq 1 + x + \alpha \, x^2 .$$

The first bound always holds; the second follows from

$$\frac{x}{3!} + \frac{x^2}{4!} + \frac{x^3}{5!} \geq \alpha - \frac{1}{2} ,$$

as the lhs is monotonically increasing and equals $-7/30$ at $x = -2$. The rhs is always smaller than $-1/4 < -7/30$. As $x_2 \geq -2$ (due to $\alpha \leq 1/4$), (29) holds for all $x \geq x_1$. $\qquad \square$

We can now lower-bound InfoNCE:

**Theorem B.1.** *Assuming that the kernel over $Z$ is bounded as $|k(z, z')| \leq k^{\max}$ for any $z, z'$, and the kernel over $Y$ satisfies $\Delta l = N$ (defined in (13)). Then for $\gamma$ satisfying $\min\{-2, -2k^{\max}\} = - \left( 1 + \sqrt{1 - 4\gamma} \right) /(2\gamma)$,*

$$- \text{HSIC}(Z, Y) + \gamma \, \text{HSIC}(Z, Z) \leq \mathcal{L}_{\text{InfoNCE}}(\theta) - \mathbb{E}_Z \frac{\left( \gamma \mathbb{V}\text{ar}_{Z'} \left[ k(Z, Z') \right] \right)^2}{1 + \gamma \mathbb{V}\text{ar}_{Z'} \left[ k(Z, Z') \right]} .$$

*Proof.* As we assumed the kernel is bounded, for $k(Z, Z') - \mu_1 \geq -2k^{\max}$ (almost surely; the factor of 2 comes from centering by $\mu_1$). Now if we choose $\gamma$ that satisfies $\min\{-2, -2k^{\max}\} = - \left( 1 + \sqrt{1 - 4\gamma} \right) /(2\gamma)$ (the minimum is to apply our bound even for $k^{\max} < 1$), then (almost surely) by Lemma B.0.1,

$$\exp \left( k(Z, Z') - \mu_1 \right) \geq 1 + k(Z, Z') - \mu_1 + \gamma \left( k(Z, Z') - \mu_1 \right)^2 .$$

Therefore, we can take the expectation w.r.t. $Z'$, and obtain

$$\mathbb{E}_Z \log \mathbb{E}_{Z'} \exp \left( k(Z, Z') - \mu_1 \right) \geq \mathbb{E}_Z \log \left( 1 + \gamma \, \mathbb{V}\text{ar}_{Z'} \left[ k(Z, Z') \right] \right) .$$

Now we can use that $\log(1 + x) \geq x/(1 + x) = x - x^2/(1 + x)$ for $x > -1$, resulting in

$$\mathbb{E}_Z \log \mathbb{E}_{Z'} \exp \left( k(Z, Z') - \mu_1 \right) \geq \gamma \, \mathbb{E}_Z \mathbb{V}\text{ar}_{Z'} \left[ k(Z, Z') \right] + \mathbb{E}_Z \frac{\left( \gamma \, \mathbb{V}\text{ar}_{Z'} \left[ k(Z, Z') \right] \right)^2}{1 + \gamma \, \mathbb{V}\text{ar}_{Z'} \left[ k(Z, Z') \right]} .$$

Using (28), we obtain that

$$\mathcal{L}_{\text{InfoNCE}}(\theta) = -\text{HSIC}(Z, Y) + \mathbb{E}_Z \log \mathbb{E}_{Z'} \exp \left( k(Z, Z') - \mu_1 \right)$$

$$\geq -\text{HSIC}(Z, Y) + \gamma \, \mathbb{E}_Z \mathbb{V}\text{ar}_{Z'} \left[ k(Z, Z') \right] + \mathbb{E}_Z \frac{\left( \gamma \mathbb{V}\text{ar}_{Z'} \left[ k(Z, Z') \right] \right)^2}{1 + \gamma \mathbb{V}\text{ar}_{Z'} \left[ k(Z, Z') \right]} .$$

Finally, noting that by Cauchy-Schwarz

$$\text{HSIC}(Z,Z) = \mathbb{E}_{Z,Z'}k(Z,Z')^2 - 2\mathbb{E}_Z\left(\mathbb{E}_{Z'}k(Z,Z')\right)^2 + \left(\mathbb{E}_{Z,Z'}k(Z,Z')\right)^2$$

$$\leq \mathbb{E}_{Z,Z'}k(Z,Z')^2 - \mathbb{E}_Z\left(\mathbb{E}_{Z'}k(Z,Z')\right)^2 = \mathbb{E}_Z\mathbb{V}\text{ar}_{Z'}\left[k(Z,Z')\right],$$

we get the desired bound. $\square$

Theorem B.1 works for any bounded kernel, because $-\left(1+\sqrt{1-4\gamma}\right)/(2\gamma)$ takes values in $(\infty, -2]$ for $\gamma \in (0, 1/4]$. For inverse temperature-scaled cosine similarity kernel $k(z, z') = z^\top z'/(\tau\|z\|\|z'\|)$, we have $k^{\max} = 1/\tau$. For $\tau = 0.1$ (used in SimCLR [19]), we get $\gamma = 0.0475$. For the Gaussian and the IMQ kernels, $k(z, z') \geq -\mu_1 \geq -1$, so we can replace $\gamma$ with $\frac{1}{3}$ due to the following inequality: for $x \geq a$,

$$\exp(x) \geq 1 + x + \frac{x^2}{2} + \frac{x^3}{6} \geq 1 + x + \frac{a+3}{6}x^2,$$

where the first inequality is always true.

## B.2  MMD interpretation of HSIC(X,Y)

The special label structure of the self-supervised setting allows to understand $\text{HSIC}(X, Y)$ in terms of the maximum mean discrepancy (MMD). Denoting labels as $i$ and $j$ and corresponding mean feature vectors (in the RKHS) as $\mu_i$ and $\mu_j$,

$$\text{MMD}^2(i,j) = \|\mu_i - \mu_j\|^2 = \langle\mu_i, \mu_i\rangle + \langle\mu_j, \mu_j\rangle - 2\langle\mu_i, \mu_j\rangle .$$

Therefore, the average over all labels becomes

$$\frac{1}{N^2}\sum_{ij}\text{MMD}^2(i,j) = \frac{2}{N}\sum_i\langle\mu_i,\mu_i\rangle - \frac{2}{N^2}\sum_{ij}\langle\mu_i,\mu_j\rangle$$

$$= \frac{2}{N}\sum_i\langle\mu_i,\mu_i\rangle - \frac{2}{N^2}\left\langle\sum_i\mu_i, \sum_j\mu_j\right\rangle$$

$$= 2\mathbb{E}_i\mathbb{E}_{Z|i,Z'|i}\langle\phi(Z),\phi(Z')\rangle - 2\left\langle\mathbb{E}_i\mathbb{E}_{Z|i}\phi(Z), \mathbb{E}_j\mathbb{E}_{Z'|j}\phi(Z')\right\rangle$$

$$= 2\mathbb{E}_{Z,Z'\sim\text{pos}}\left[k(Z,Z')\right] - 2\mathbb{E}_Z\mathbb{E}_{Z'}\left[k(Z,Z')\right],$$

where the second last line uses that all labels have the same probability $1/N$, and the last line takes the expectation out of the dot product and uses $k(Z,Z') = \langle\phi(Z), \phi(Z')\rangle$.

Therefore,

$$\frac{1}{2N^2}\sum_{ij}\text{MMD}^2(i,j) = \frac{N}{\Delta l}\text{HSIC}(Z,Y).$$

## B.3  Centered representation assumption for clustering

In Section 3.2, we make the assumption that the features are centered and argue that the assumption is valid for BYOL. Here we show empirical evidence of centered features. First, we train BYOL for 1000 epochs which reaches a top-1 accuracy of 74.5% similar to the reported result in [25]. Next, we extract feature representations (predictor and target projector outputs followed by re-normalization) under training data augmentations for a batch of 4096 images. One sample Z-test is carried out on the feature representations with $H_0 : \mu = 0$ and $H_1 : \mu \neq 0$. The null hypothesis is accepted under threshold $\alpha = 0.025$.

# C  Random Fourier Features (RFF)

## C.1  Basics of RFF

Random Fourier features were introduced by Rahimi and Recht [33] to reduce computational complexity of kernel methods. Briefly, for translation-invariant kernels $k(z - z')$ that satisfy $k(0) = 1$,

Bochner's theorem gives that

$$k(z - z') = \int p(\omega) e^{i\omega^\top (z - z')} d^n\omega = \mathbb{E}_\omega \left[ e^{i\omega^\top z} \left( e^{i\omega^\top z'} \right)^* \right] ,$$

where the probability distribution $p(\omega)$ is the $n$-dimensional Fourier transform of $k(z - z')$.

As both the kernel and $p(\omega)$ are real-valued, we only need the real parts of the exponent. Therefore, for $b \sim \text{Uniform}[0, 2\pi]$,

$$k(z - z') = \mathbb{E}_{\omega, b} \left[ 2 \cos(\omega^\top z + b) \cos(\omega^\top z' + b) \right] .$$

For $N$ data points, we can draw $D$ $\omega_d$ from $p(\omega)$, construct RFF for each points $z_i$, put them into matrix $R \in \mathbb{R}^{N \times D}$, and approximate the kernel matrix as

$$K \approx RR^\top, \ R_{id} = \sqrt{\frac{2}{D}} \cos(\omega_d^\top z_i + b) ,$$

and $\mathbb{E} \, RR^\top = K$.

For the Gaussian kernel, $k(z - z') = \exp(-\|z - z'\|^2/2)$, we have $p(\omega) = (2\pi)^{-D/2} \exp(-\|\omega\|^2/2)$ [33]. We are not aware of literature on RFF representation for the inverse multiquadratic (IMQ) kernel; we derive it below using standard methods.

## C.2   RFF for the IMQ kernel

**Theorem C.1.** *For the inverse multiquadratic (IMQ) kernel,*

$$k(z, z') \equiv k(z - z') = \frac{c}{\sqrt{c^2 + \|z - z'\|^2}} ,$$

*the distribution of random Fourier features $p(\omega)$ is proportional to the following (for $s = \|w\|$),*

$$p(\omega) \equiv \hat{h}(s) \propto \frac{K_{\frac{n-2}{2} + \frac{1}{2}}(cs)}{s^{\frac{n-2}{2} + \frac{1}{2}}} = \frac{K_{\frac{n-1}{2}}(cs)}{s^{\frac{n-1}{2}}} , \tag{30}$$

*where $K_\nu$ is the modified Bessel function (of the second kind) of order $\nu$.*

*Proof.* To find the random Fourier features, we need to take the Fourier transform of this kernel,

$$\hat{k}(\omega) = \int e^{-i\omega^\top z} k(z) d^n z .$$

As the IMQ kernel is radially symmetric, meaning that $k(z, z') = h(r)$ for $r = \|z - z'\|$, its Fourier transform can be written in terms of the Hankel transform [73, Section B.5] (with $\|\omega\| = s$)

$$\hat{k}(\omega) = \hat{h}(s) = \frac{(2\pi)^{n/2}}{s^{\frac{n-2}{2}}} H_{\frac{n-2}{2}} \left[ r^{\frac{n-2}{2}} h(r) \right] (s) .$$

The Hankel transform of order $\nu$ is defined as

$$H_\nu[g(t)](s) = \int_0^\infty J_\nu(st) g(t) t \, dt ,$$

where $J_\nu(s)$ is the Bessel function (of the first kind) of order $\nu$.

As $h(r) = c/\sqrt{c^2 + r^2}$,

$$H_{\frac{n-2}{2}} \left[ r^{\frac{n-2}{2}} h(r) \right] (s) = c \frac{\sqrt{2} c^{\frac{n-2}{2} + 1/2}}{\sqrt{s} \Gamma(\frac{1}{2})} K_{\frac{n-2}{2} + \frac{1}{2}}(cs) ,$$

where $K_\nu$ is a modified Bessel function (of the second kind) of order $\nu$.

Therefore, by using a table of Hankel transforms [74, Chapter 9, Table 9.2],

$$\hat{h}(s) \propto \frac{K_{\frac{n-2}{2} + \frac{1}{2}}(cs)}{s^{\frac{n-2}{2} + \frac{1}{2}}} = \frac{K_{\frac{n-1}{2}}(cs)}{s^{\frac{n-1}{2}}} .$$

$\square$

### C.2.1 How to sample

To sample random vectors from (30), we can first sample their directions as uniformly distributed unit vectors $d/\|d\|$, and then their amplitudes $s$ from $\hat{h}(s)s^{n-1}$ (the multiplier comes from the change to spherical coordinates).

Sampling unit vectors is easy, as for $d \sim \mathcal{N}(0, I)$, $d/\|d\|$ is a uniformly distributed unit vector.

To sample the amplitudes, we numerically evaluate

$$\tilde{p}(s) = \hat{h}(s)s^{n-1} = K_{\frac{n-1}{2}}(cs)s^{\frac{n-1}{2}} \tag{31}$$

on a grid, normalize it to get a valid probability distribution, and sample from this approximation. As for large orders $K_\nu$ attains very large numbers, we use mpmath [75], an arbitrary precision floating-point arithmetic library for Python. As we only need to sample $\tilde{p}(s)$ once during training, this adds a negligible computational overhead.

Finally, note that for any IMQ bias $c$, we can sample $s$ from (31) for $c = 1$, and then use $\tilde{s} = s/c$ to rescale the amplitudes. This is because

$$P(s/c \le x) = P(s \le cx) = C \int_0^{cx} K_{\frac{n-1}{2}}(t)t^{\frac{n-1}{2}} dt = Cc^{\frac{n-1}{2}} \int_0^x K_{\frac{n-1}{2}}(c\tilde{t})\tilde{t}^{\frac{n-1}{2}} cd\tilde{t}.$$

In practice, we evaluate $\tilde{p}(s)$ for $c = 1$ on a uniform grid over $[10^{-12}, 100]$ with $10^4$ points, and rescale for other $c$ (for output dimensions of more that 128, we use a larger grid; see Appendix E).

### C.3 RFF for SSL-HSIC

To apply RFF to SSL-HSIC, we will discuss the $\text{HSIC}(Z, Y)$ and the $\text{HSIC}(Z, Z)$ terms separately. We will use the following notation:

$$k(z_i^p, z_j^l) \approx \sum_{d=1}^D r_d^{ip} r_d^{jl}$$

for $D$-dimensional RFF $r^{ip}$ and $r^{jl}$.

Starting with the first term, we can re-write (17) as

$$\widehat{\text{HSIC}}(Z, Y)_{\text{RFF}} = \frac{1}{BM(M-1)} \sum_{ipld} r_d^{ip} r_d^{il} - \frac{1}{B^2 M^2} \sum_{ijpld} r_d^{ip} r_d^{jl} - \frac{1}{M-1}$$

$$= \frac{1}{BM(M-1)} \sum_{id} \left( \sum_p r_d^{ip} \right)^2 - \frac{1}{B^2 M^2} \sum_d \left( \sum_{ip} r_d^{ip} \right)^2 - \frac{1}{M-1}.$$

The last term in the equation above is why we use RFF: instead of computing $\sum_{ijpl}$ in $O(B^2 M^2)$ operations, we compute $\sum_{ip}$ in $O(BM)$ and then sum over $d$, resulting in $O(BMD)$ operations (as we use large batches, typically $BM > D$). As $\text{HSIC}(Z, Y)$ is linear in $k$, $\mathbb{E}_{\omega,b} \widehat{\text{HSIC}}(Z, Y)_{\text{RFF}} = \widehat{\text{HSIC}}(Z, Y)$.

To estimate $\widehat{\text{HSIC}}(Z, Z)$, we need to sample RFF twice. This is because

$$\widehat{\text{HSIC}}(Z, Z) = \frac{1}{(BM-1)^2} \text{Tr}(KHKH),$$

therefore we need the first $K$ to be approximated by $RR^\top$, and the second – by an independently sampled $\tilde{R}\tilde{R}^\top$. This way, we will have $\mathbb{E}_{\omega,b,\tilde{\omega},\tilde{b}} \widehat{\text{HSIC}}(Z, Z)_{\text{RFF}} = \widehat{\text{HSIC}}(Z, Z)$.

Therefore, we have (noting that $HH = H$)

$$
\begin{aligned}
\widehat{\text{HSIC}}(Z, Z)_{\text{RFF}} &= \frac{1}{(BM-1)^2} \text{Tr}\left(RR^\top H \tilde{R}\tilde{R}^\top H\right) = \frac{1}{(BM-1)^2}\|R^\top H \tilde{R}\|_F^2 \\
&= \frac{1}{(BM-1)^2}\|R^\top HH \tilde{R}\|_F^2 \\
&= \frac{1}{(BM-1)^2}\sum_{d_1, d_2}\left(\sum_{ip}\left(r_{d_1}^{ip} - \frac{1}{BM}\sum_{jl}r_{d_1}^{jl}\right)\left(r_{d_2}^{ip} - \frac{1}{BM}\sum_{jl}r_{d_2}^{jl}\right)\right)^2.
\end{aligned}
$$

To summarize the computational complexity of this approach, computing $D$ random Fourier features for a $Q$-dimensional $z$ takes $O(DQ)$ operations (sampling $D \times K$ Gaussian vector, normalizing it, sampling $D$ amplitudes, computing $\omega_d^\top z$ $D$ times), therefore $O(BMDQ)$ for $BM$ points. After that, computing $\text{HSIC}(Z, Y)$ takes $O(BMD)$ operations, and $\text{HSIC}(Z, Z) - O(BMD^2)$ operations. The resulting complexity per batch is $O(BMD(Q + D))$. Note that we sample new features every batch.

In contrast, computing SSL-HSIC directly would cost $O(Q)$ operations per entry of $K$, resulting in $O((BM)^2Q)$ operations. Computing HSIC would then be quadratic in batch size, and the total complexity would stay $O((BM)^2Q)$.

In the majority of experiments, $B = 4096$, $M = 2$, $Q = 128$ and $D = 512$, and the RFF approximation performs faster (with little change in accuracy; see Table 8b).

# D   Experiment Details

## D.1   ImageNet Pretraining

### D.1.1   Data augmentation

We follow the same data augmentation scheme as BYOL [25] with exactly the same parameters. For completeness, we list the augmentations applied and parameters used:

- random cropping: randomly sample an area of $8\%$ to $100\%$ of the original image with an aspect ratio logarithmically sampled from $3/4$ to $4/3$. The cropped image is resized to $224 \times 224$ with bicubic interpolation;

- flip: optionally flip the image with a probability of $0.5$;

- color jittering: adjusting brightness, contrast, saturation and hue in a random order with probabilities $0.8$, $0.4$, $0.4$, $0.2$ and $0.1$ respectively;

- color dropping: optionally converting to grayscale with a probability of $0.2$;

- Gaussian blurring: Gaussian kernel of size $23 \times 23$ with a standard deviation uniformly sampled over $[0.1, 2.0]$;

- solarization: optionally apply color transformation $x \mapsto x \cdot 1_{x<0.5} + (1 - x) \cdot 1_{x \geq 0.5}$ for pixels with values in $[0, 1]$. Solarization is only applied for the second view, with a probability of $0.2$.

### D.1.2 Optimizing kernel parameters

Since we use radial basis function kernels, we can express the kernel $k(s)$ in term of the distance $s = \|z_i - z_j\|^2$. The entropy of the kernel distance $k_\sigma(s_{ij})$ can be expressed as follows:

$$
\begin{aligned}
H[k] &= -\int p(k)\log p(k)dk \\
&= -\int q(s)\log\left(q(s)\left|\frac{ds}{dk}\right|\right)ds \\
&= H[s] + \int q(s)\log\left|\frac{dk}{ds}\right|ds \\
&= \mathbb{E}\left[\log|k_\sigma'(s)|\right] + \text{const} \\
&\propto \mathbb{E}\left[\log|k_\sigma'(s)|^2\right] + \text{const}.
\end{aligned}
$$

We use the kernel distance entropy to automatically tune kernel parameters: for the kernel parameter $\sigma$, we update it to maximize $\mathbb{E}\left[\log|k_\sigma'|^2\right]$ (for IMQ, we optimize the bias $c$) at every batch. This procedure makes sure the kernel remains sensitive to data variations as representations move closer to each other.

## D.2 Evaluations

### D.2.1 ImageNet linear evaluation protocol

After pretraining with SSL-HSIC, we retain the encoder weights and train a linear layer on top of the frozen representation. The original ImageNet training set is split into a training set and a local validation set with 10000 data points. We train the linear layer on the training set. Spatial augmentations are applied during training, i.e., random crops with resizing to $224 \times 224$ pixels, and random flips. For validation, images are resized to 256 pixels along the shorter side using bicubic resampling, after which a $224 \times 224$ center crop is applied. We use `SGD` with Nesterov momentum and train over 90 epochs, with a batch size of 4096 and a momentum of 0.9. We sweep over learning rate and weight decay and choose the hyperparameter with top-1 accuracy on local validation set. With the best hyperparameter setting, we report the final performance on the original ImageNet validation set.

### D.2.2 ImageNet semi-supervised learning protocol

We use ImageNet 1% and 10% datasets as SimCLR [19]. During training, we initialize the weights to the pretrained weights, then fine-tune them on the ImageNet subsets. We use the same training procedure for augmentation and optimization as the linear evaluation protocol.

### D.2.3 Linear evaluation protocol for other classification datasets

We use the same dataset splits and follow the same procedure as BYOL [25] to evaluate classification performance on other datasets, i.e. 12 natural image datasets and Pascal VOC 2007. The frozen features are extracted from the frozen encoder. We learn a linear layer using logistic regression in `sklearn` with l2 penalty and `LBFGS` for optimization. We use the same local validation set as BYOL [25] and tune hyperparameter on this local validation set. Then, we train on the full training set using the chosen weight of the l2 penalty and report the final result on the test set.

### D.2.4 Fine-tuning protocol for other classification datasets

Using the same dataset splits described in Appendix D.2.3, we initialize the weights of the network to the pretrained weights and fine-tune on various classification tasks. The network is trained using `SGD` with Nestrov momentum for 20000 steps. The momentum parameter for the batch normalization statistics is set to $\max(1 - 10/s, 0.9)$ where $s$ is the number of steps per epoch. We sweep the weight decay and learning rate, and choose hyperparameters that give the best score on the local validation set. Then we use the selected weight decay and learning rate to train on the whole training set to report the test set performance.

### D.2.5 Transfer to semantic segmentation

In semantic segmentation, the goal is to classify each pixel. The head architecture is a fully-convolutional network (FCN)-based [76] architecture as [17, 25]. We train on the `train_aug2012` set and report results on `val2012`. Hyperparameters are selected on a 2119 images, which is the same held-out validation set as [25]. A standard per-pixel softmax cross-entropy loss is used to train the FCN. Training uses random scaling (by a ratio in $[0.5, 2.0]$), cropping (crop size 513), and horizontal flipping for data augmentation. Testing is performed on the $[513, 513]$ central crop. We train for 30000 steps with a batch size of 16 and weight decay $10^{-4}$. We sweep the base learning rate with local validation set. We use the best learning rate to train on the whole training set and report on the test set. During training, the learning rate is multiplied by $0.1$ at the $70th$ and $90th$ percentile of training. The final result is reported with the average of 5 seeds.

### D.2.6 Transfer to depth estimation

The network is trained to predict the depth map of a given scene. We use the same setup as BYOL [25] and report it here for completeness. The architecture is composed of a ResNet-50 backbone and a task head which takes the $conv5$ features into 4 upsampling blocks with respective filter sizes 512, 256, 128, and 64. Reverse Huber loss function is used for training. The frames are down-sampled from $[640, 480]$ by a factor 0.5 and center-cropped to size $[304, 228]$. Images are randomly flipped and color transformations are applied: greyscale with a probability of 0.3; brightness adjustment with a maximum difference of 0.1255; saturation with a saturation factor randomly picked in the interval $[0.5, 1.5]$; hue adjustment with a factor randomly picked in the interval $[-0.2, 0.2]$. We train for 7500 steps with batch size 256, weight decay 0.001, and learning rate 0.05.

### D.2.7 Transfer to object detection

We follow the same setup for evaluating COCO object detection tasks as in DetCon [77]. The architecture used is a Mask-RCNN [78] with feature pyramid networks [79]. During training, the images are randomly flipped and resized to $(1024 \cdot s) \times (1024 \cdot s)$ where $s \in [0.8, 1.25]$. Then the resized image is cropped or padded to a $1024 \times 1024$. We fine-tune the model for 12 epochs ($1\times$ schedule [17]) with `SGD` with momentum with a learning rate of 0.3 and momumtem 0.9. The learning rate increases linearly for the first 500 iterations and drops twice by a factor of 10, after $2/3$ and $8/9$ of the total training time. We apply a weight decay of $4 \times 10^{-5}$ and train with a batch size of 64.

## E  SSL-HSIC pseudo-code

```python
import jax
import jax.numpy as jnp
import mpmath
import numpy as np

def ssl_hsic_loss(hiddens, kernel_param, num_rff_features, gamma, rng):
  """Compute SSL-HSIC loss."""
  hsic_yz = compute_hsic_yz(hiddens, num_rff_features, kernel_param, rng)
  hsic_zz = compute_hsic_zz(hiddens, num_rff_features, kernel_param, rng)
  return - hsic_yz + gamma * jnp.sqrt(hsic_zz)

def compute_hsic_yz(hiddens, num_rff_features, kernel_param, rng):
  """Compute RFF approximation of HSIC_YZ."""
  # B - batch size; M - Number of transformations.
  B = hiddens[0].shape[0]
  M = len(hiddens)

  rff_hiddens = jnp.zeros((B, num_rff_features))
  mean = jnp.zeros((1, num_rff_features))
  for hidden in hiddens:
    rff_features = imq_rff_features(hidden, num_rff_features, kernel_param, rng)
    rff_hiddens += rff_features
    mean += rff_features.sum(0, keepdims=True)
  return (rff_hiddens ** 2).sum() / (B * K * (K - 1)) - (mean ** 2).sum() / (B * M) ** 2

def compute_hsic_zz(hiddens, num_rff_features, kernel_param, rng):
  """Compute RFF approximation of HSIC_ZZ."""
  rng_1, rng_2 = jax.random.split(rng, num=2)
  B = hiddens[0].shape[0]
```

```python
    M = len(hiddens)

    z1_rffs = []
    z2_rffs = []
    center_z1 = jnp.zeros((1, num_rff_features))
    center_z2 = jnp.zeros((1, num_rff_features))
    for hidden in hiddens:
        z1_rff = imq_rff_features(hidden, num_rff_features, kernel_param, rng_1)
        z2_rff = imq_rff_features(hidden, num_rff_features, kernel_param, rng_2)
        z1_rffs.append(z1_rff)
        center_z1 += z1_rff.mean(0, keepdims=True)
        z2_rffs.append(z2_rff)
        center_z2 += z2_rff.mean(0, keepdims=True)
    center_z1 /= M
    center_z2 /= M

    z = jnp.zeros(shape=(num_rff_features, num_rff_features), dtype=jnp.float32)
    for z1_rff, z2_rff in zip(z1_rffs, z2_rffs):
        z += jnp.einsum('ni,nj->ij', z1_rff - center_z1, z2_rff - center_z2)
    return (z ** 2).sum() / (B * M - 1) ** 2

def imq_rff_features(hidden, num_rff_features, kernel_param, rng):
    """Random Fourier features of IMQ kernel."""
    d = hidden.shape[-1]
    rng1, rng2 = jax.random.split(rng)
    amp, amp_probs = amplitude_frequency_and_probs(d)
    amplitudes = jax.random.choice(rng1, amp, shape=[num_rff_features, 1], p=amp_probs)
    directions = jax.random.normal(rng2, shape=(num_rff_features, d))
    b = jax.random.uniform(rng2, shape=(1, num_features)) * 2 * jnp.pi
    w = directions / jnp.linalg.norm(directions, axis=-1, keepdims=True) * amplitudes
    z = jnp.sqrt(2 / num_rff_features) * jnp.cos(jnp.matmul(hidden / kernel_param, w.T) + b)
    return z

def amplitude_frequency_and_probs(d):
    """Returns frequencies and probabilities of amplitude for RFF of IMQ kernel."""
    # Heuristics for increasing the upper limit with the feature dimension.
    if d >= 4096:
        upper = 200
    elif d >= 2048:
        upper = 150
    elif d >= 1024:
        upper = 120
    else:
        upper = 100
    x = np.linspace(1e-12, upper, 10000)
    p = compute_prob(d, x)
    return x, p

def compute_prob(d, x_range):
    """Returns probabilities associated with the frequencies."""
    prob = list()
    for x in x_range:
        prob.append(mpmath.besselk((d - 1) / 2, x) * mpmath.power(x, (d - 1) / 2))
    normalizer = prob[0]
    for x in prob[1:]:
        normalizer += x
    normalized_prob = []
    for x in prob:
        normalized_prob.append(float(x / normalizer))
    return np.array(normalized_prob)
```