# OpenReview forum: "Self-Supervised Learning with Kernel Dependence Maximization"
_NeurIPS.cc/2021/Conference — NeurIPS 2021 Poster_

### Official Review · Reviewer_j3wZ · 2021-07-14

**Rating:** 5
**Confidence:** 4

**Summary:**


This paper proposed a new loss function for self-supervised learning using the Hilbert-Schmidt Independence Criterion (HSIC) to measure the kernel space dependence between the learned latent representation of images and their augmented versions. The main contribution is that authors demonstrated the connections between the proposed framework and InfoNCE, metric learning and Maximum Mean Discrepancy, respectively. An algorithm with random fourier feature kernel estimation is proposed and the method is shown to have comparable performance with state-of-the-art self-supervised learning (SSL) paradigms. The sample complexity and bias analysis for kernel estimation are provided.


**Limitations And Societal Impact:**

none noted

**Main Review:**

Strengths:
- This paper offers an interesting viewpoint and an alternative formulation for maximal mutual information-based self-supervised learning (SSL). It makes much intuitive sense that one could maximize the minus HSIC to maximize the dependence between two random variables. The proposed method has some advantages in terms of measuring the gap between the population formulation and its sample based ones, reminiscent of HSIC and kernel estimation results. The combination of HSIC and random feature-based kernel computation is natural. The sample complexity analysis supporting this part makes sense. The regularization also makes sense, which is essentially a norm regularization in the kernel space.
- The paper is relatively well written, although the introduction of Y (image identify) may be confusing.


Weaknesses and questions:
- The novelty seems to be limited and impact in terms of changing the SSL paradigms could be marginal.  Essentially, the proposed method follows [35] but changes to another mutual information (MI) approximator. The effectiveness of this approximator is not rigorously shown but argued from a bit of a loose angle. In particular, the authors argued that exact MI maximization may not be desired due to invertible ambiguities. But it is hard to see why using a looser lower bound is a better choice. In other words, the motivation of using HSIC to replace MI is not crystal clear.  It indeed makes sense that maximizing dependence of different views may not give the best result of SSL. But it feels not easy to grasp the idea of intentionally using a loose lower bound for MI. Why is a loose lower bound better? How loose should one aim for? This is quite unclear to the reviewer. One suspicion is that maybe a good regularization (like the HSIC(Z,Z) regularization) can fix the ambiguity brought by the dependence criterion. How much performance gain is brought upon by the –HSIC part? Ablation study on $\gamma$ could perhaps provide some insights.
- The connections between the proposed regularized HSIC and InfoNCE criteria are somewhat interesting, but not identical or equivalent. It does not feel that such a loose connection merits a new publication.
- Using the kernel method may make the computation a bit more challenging. Indeed, random features could alleviate the memory burden, but it introduces another level of approximation. This also goes back to the previous comment - there seems to be no strong motivation for using HSIC.

- Following the above comment, what is the unique or salient feature of this entire SSL framework is unclear. As a benefit of the proposed method mentioned by the authors, large negative sampling is not needed. However, to the reviewer’s understanding, [23] also has this advantage, which interprets contrastive loss as alignment and uniformity. How is the performance of the proposed method compared with [23]?
- The proposed formulation seems confusing. In Eq. (4) Y is used, but the actual estimation for HSIC(Z,Y) does not require the information of Y (otherwise it is supervised rather than self-supervised). This was quite confusing when first reading the paper, since SSL does not need the label information Y. By looking into the supplementary material, it was then clever that essentially the kernel dependence maximization is computed between the sample and its augmented counterpart. The Y information was never used in computation. Perhaps it would be better and clearer to elaborate on this gap between the conceptual formulation and the practical implementation. Is it needed to introduce the “label” or image identity term y when formulating the problem? Removing it may make the story smoother, in the reviewer’s opinion.
-  The analyses mostly are concerned with sample complexity and kernel estimation accuracy. These are of course not uninteresting, but are not exactly relevant to the main question, i.e., how the HSIC formulation affects SSL?
- The experiment results do not really show improvement compared with state-of-the-art in most cases. For example, in transfer learning tasks (Table 3), most results are not improved compared to other approaches, even the better ones are only with very small margins. Similar results are also observed in segmentation (Table 4) and object detection (Table 5) tasks. Besides, as mentioned above, study on gamma could be interesting and helpful to evaluate the contribution of the regularization term, i.e., HSIC(Z,Z). Overall, it seems that the proposed method does not make a big difference in terms of empirical performances.
- On the computational side, it seems to require more computations when using kernel functions even though random Fourier features are employed. Is there a time consumption comparison with existing methods to illustrate the computation cost?

- There are some discussions on scaling the HSIC estimator to reduce bias, and some analyses to support the design. How much does the scaling impact the final result? Could this be validated or supported by some simulations? This part also reads a bit confusing. From the paragraph, it seems that the goal is to handle non-iid samples. But the proofs in Supp Sec. A seem to still use iid assumptions. Can the authors clarify a bit?
- Overall, the HSIC based dependence measuring seems to be reasonable, but its motivation and impacts in practice are not crystal clear.







**Time Spent Reviewing:**

3

---

> ### Author Response · Authors · 2021-08-10
> **Official response**
>
> Thank you for the review! Below we try to address each of your concerns; if any issues remain, we’d be very happy to discuss further.
> > the motivation of using HSIC to replace MI is not crystal clear.
>
> First, we would like to convince the reviewer that MI is not a good measure to optimize for representation learning.
>
> Here’s a simple explicit example where many possible representations have the same MI, but drastically different representation qualities. Consider a problem with two inputs, $A$ and $B$, and a one-dimensional featurizer, parameterized by the integer $M$, which maps $A$ to $\mathrm{Uniform}(\\{0, 2, \dots, 2M\\}) + \mathrm{Uniform}((-0.5, 0.5))$ and $B$ to $\mathrm{Uniform}(\\{1, 3, \dots, 2M+1\\}) + \mathrm{Uniform}((-0.5, 0.5))$. When $M = 0$, the two inputs are simply separated into $A \in (-0.5, 0.5)$ and $B \in (0.5, 1.5)$. But if $M$ is large, they are interspersed like ABABABAB – a representation which is clearly much harder to work with for most downstream learners. Nevertheless, the mutual information between the features of any two augmentations of the same input (a positive pair) is independent of $M$, simply $H[Z_1] - H[Z_1 \mid Z_2] = \log 2$ for any $M$. HSIC with any reasonable kernel choice, by contrast, would strongly prefer $M = 0$.
>
> As pointed out by https://arxiv.org/abs/1907.13625 [33], the success of InfoNCE cannot be attributed to the properties of mutual information, but depends on the actual form of the estimators used – that is, the lower bounds behave very differently from the MI itself. In fact, Section 3 of [33] shows (in a more practical version of our example above) that the features can be very poor with an encoder that provably maximizes MI. In other words, the quality of the learned features is not correlated to how good the estimate of mutual information is.
>
> In general, mutual information is invariant to smooth and invertible transformations of random variables. Therefore, MI doesn’t really care about the geometry of representations, while the InfoNCE estimator does. This is an important factor that made it work for SSL. We argue that HSIC is a dependency measure that cares about geometry of representations by definition, and it is a better way to understand InfoNCE and even BYOL. It is also easy to estimate directly with little bias, meaning that we know exactly what our loss is estimating and what our representation is optimized for. (Please also see our comment to all reviewers for more about the motivation for using HSIC.)
> > Essentially, the proposed method follows [35] but changes to another mutual information (MI) approximator.
>
> As explained in the response to all reviewers, HSIC is not another MI approximator, but a fundamentally different notion of dependence that is both better-motivated and more practical.
> > The effectiveness of this approximator is not rigorously shown but argued from a bit of a loose angle.
>
> The effectiveness of SSL-HSIC is shown both theoretically via relations to existing successful SSL methods and empirically with competitive performance across all the tasks.
> > maybe a good regularization (like the HSIC(Z,Z) regularization) can fix the ambiguity brought by the dependence criterion
>
> Regularization is important for good performance; not having any leads to a failure mode where the intermediate features get very large. However, different forms of regularization could work, such as the implicit one in InfoNCE. We do show that $HSIC(Z,Z)$ is a better regularizer than InfoNCE in Table 6a. (Additionally, the HSIC regularizer allows us to use random Fourier features for better scalability). In general, we agree that the regularizer should help eliminate the ambiguity brought by $HSIC(Z, Y)$, as this part doesn’t change if you reshuffle clusters belonging to each original image.
> > The connections between the proposed regularized HSIC and InfoNCE criteria are somewhat interesting, but not identical or equivalent.
>
> As argued in the response to all reviewers, we claim that the success of InfoNCE is not because it is a lower bound to MI, but rather because of its relation to SSL-HSIC. We show that InfoNCE is an HSIC-based loss with a specific regularizer, and that it tightly approximates $-HSIC(Z, Y) + \gamma HSIC(Z, Z)$. We propose a slightly better HSIC-based loss, with a square root on the regularizer; this works somewhat better empirically, and makes more sense from first principles as discussed briefly in footnote 2. We will expand this discussion and make the connection clearer in revision.
> > Using the kernel method may make the computation a bit more challenging.
>
> With RFF, our method scales linearly to the batch size. Compared to SSL methods that use negative samples which scale quadratic to the batch size, we have a better computational complexity. Compared to non-negative methods which also scale linearly to the batch size, our method has a larger overhead depending on the number of Fourier features used. We didn’t run an exact wall clock time comparison for the computational cost, but our experiments weren’t significantly slower in practice than BYOL.
> > Following the above comment, what is the unique or salient feature of this entire SSL framework is unclear.
>
> Please see our comments to all the reviewers regarding SSL methods and their performance, as well as the discussion above.
> > As a benefit of the proposed method mentioned by the authors, large negative sampling is not needed. However, to the reviewer’s understanding, [23] also has this advantage, which interprets contrastive loss as alignment and uniformity. How is the performance of the proposed method compared with [23]?
>
> From the limited large-scale experiment provided by [23] (200 epochs on ImageNet), the performance of downstream ImageNet classification top-1 accuracy is 67.69%. Our result on the same downstream task after 1000 epochs pretraining is 74.8%, much stronger performance than [23]. On the number of negative examples, [23] uses the MoCo framework with a queue size of 65536 for the ImageNet experiment. That is effectively using negative samples in the order of 65k, which in our opinion doesn’t support the claim that [23] has the property of degrading gracefully with decreasing negative sample size. There are other SSL methods, such as BYOL, that have this property, but they don’t have a well-understood theoretical framework to support any analysis or improvements.
> > The proposed formulation seems confusing. [...] Is it needed to introduce the “label” or image identity term y when formulating the problem?
>
> We can understand that the self-supervised label Y could be a point of confusion. We never use downstream label information (e.g. ImageNet class) in training our model, but use a self-supervised label Y for image identity. This is of the same nature as the “label” in InfoNCE. InfoNCE uses Noise Contrastive Estimator (NCE) first discussed by https://proceedings.mlr.press/v9/gutmann10a/gutmann10a.pdf. Section 2.2 of that paper lays out the connection to supervised learning. Effectively, we could see NCE as assigning a “label” to a set with elements coming from both the target and noise distributions. It is still an unsupervised method since the “label” is not assigned by any human annotator. We use self-supervised labels throughout our paper to refer to this type of “label” (which we also call image identity). We think that starting from the self-supervised label helps us better understand what the target cluster structure is for SSL-HSIC. We will provide a clearer explanation in the next revision of the paper to emphasize what we mean by a “label” here.
> > The analyses mostly are concerned with sample complexity and kernel estimation accuracy. These are of course not uninteresting, but are not exactly relevant to the main question, i.e., how the HSIC formulation affects SSL?
> >
> > The experiment results do not really show improvement compared with state-of-the-art in most cases. [...] Overall, it seems that the proposed method does not make a big difference in terms of empirical performances.
>
> Please see our comments to all the reviewers regarding SSL methods and their performance.
> > Besides, as mentioned above, study on gamma could be interesting and helpful to evaluate the contribution of the regularization term, i.e., HSIC(Z,Z).
>
> Please see our comments about regularization above.
> > On the computational side, it seems to require more computations when using kernel functions even though random Fourier features are employed.
>
> Please see our comments above about computational cost.
> > There are some discussions on scaling the HSIC estimator to reduce bias, and some analyses to support the design. [...] From the paragraph, it seems that the goal is to handle non-iid samples. But the proofs in Supp Sec. A seem to still use iid assumptions.
>
> Sampling a single batch from the training dataset is i.i.d. Dependence is introduced when we apply transformations of the images. After applying the transformations, a single batch of data consists of multiple transformations of the same image.
>
> In principle, the scaling factor could be absorbed by gamma. If we tune the hyperparameter gamma correctly, the final performance should be unaffected. However, the biased HSIC(Z,Y) typically takes a small value, affecting the optimization process and accuracy of the gradient. This can be fixed by using a global scaling factor for the loss, but we think such formulation is more confusing.
> > Overall, the HSIC based dependence measuring seems to be reasonable, but its motivation and impacts in practice are not crystal clear.
>
> We hope we have shown that SSL-HSIC is not only a reasonable measure, but a measure with a clearly understood supporting framework. This framework helps us to explain the properties of successful SSL methods. Possessing the same properties, SSL-HCIS performs competitively to the state-of-the-art. This framework could be used to help with future research directions.

---

### Official Review · Reviewer_dxjH · 2021-07-16

**Rating:** 8
**Confidence:** 4

**Summary:**

Authors proposes a novel HSIC based self-supervised learning loss for image representation learning. It relates to InfoNCE and Metric Learning and provides representations that perform well on multiple tasks.

**Ethical Concerns:**

No specific ethical concerns.

**Limitations And Societal Impact:**

Limitations of the technique are addressed but bias and explainability aspects of learned representations are not, even qualitatively.

**Main Review:**

Paper introduces HSIC based SSL loss that learns generic image representations that perform well across multiple image related tasks. Strong empirical results on ImageNet classification and other tasks such as semantic segmentation, depth estimation and object detection.  It makes a good case where generic learned representations valuables as task specific representations will do better in terms of performance.

Framework uses unique identity of the image as part of loss function (y) which is a strong assumption but authors do describe it as an assumption which can be addressed.

Paper is well written with strong theoretical support of the intuition behind SSL-HSIC loss with ablation studies and choice of parameters.

A t-SNE visualizations of SSL-HSIC with class label would provide a qualitative analysis of the learn representations. It would be interesting to know if it helps improve  Top-1% accuracy of imageNet where SwaV does better (Table 1).



**Needs Ethics Review:**

Yes

**Time Spent Reviewing:**

4

---

> ### Author Response · Authors · 2021-08-10
> **Official response**
>
> Thank you for your review! Our responses are below - if any issues remain, we’d be very happy to discuss further.
>
> 1. A t-SNE analysis would indeed be interesting, and we will look into that for the updated version. One would hope to see meaningful differences between approaches (especially when using/not using a target network). However, it might not be sensitive enough to pick up a few percent accuracy changes for something as large as ImageNet.
>
> 2. We didn’t quite understand what you mean by improvements through t-SNE. Are you thinking about reducing the dimensionality with t-SNE, and then clustering the resulting representations for a harder label assignment? That would indeed be an interesting thing to try, but perhaps in an independent paper.

---

### Official Review · Reviewer_ArQY · 2021-07-16

**Rating:** 6
**Confidence:** 5

**Summary:**

The paper proposes to use the HSIC metric in order to impose view feature dependency on the image identity during self-supervised learning (SSL). The authors made non-trivial efforts in both deriving the SSL-HSIC approach and theoretically linking SSL-HSIC with the existing mainstream SSL approaches. Empirical results show that the method offers competitive results.

**Limitations And Societal Impact:**

Yes, the authors adequately addressed the limitations and potential negative societal impact of their work.

**Main Review:**

Strength: The paper is well written, and the paper is fairly easy to follow. The intuition behind the method is clear and convincing: HSIC regularizer explicitly penalizes the dependency between different images while it encourages views from the same images to be close. The theoretical results are strong.

Weakness:
1. The eventual loss boils down to Eq. (11). Particualrly, if one employs Gaussian Kernel on the eventual loss on Eq. (11), it is immediate that the first term in Eq.(5)/Eq. (11) is proportional to BYOL up to a scaler, whereas the second term further plays additional contrastive role in the form of inter-image $\ell_2$ norm on top of the first term; In this sense, this extra contrastive term perhaps explains why SSL-HSIC always remains comparable to BYOL and behaves potentially superior over BYOL. However, my main concern is that, given that SSL-HSIC is closely related to BYOL, it seems the proposed approach SSL-HSIC is still inferior to the BYOL in most of the experiments shown in both Table 2 and 3. Is there any insight behind this phenomenon?  Intuitivelly, SSL_HSIC poses tighter constraints based on this extra contrastive term. Also, for the same reason, the empirical results failed to advacate on the SSL-HSIC side in comparison to BYOL, making the empirical justification much weaker for SSL-HSIC.  Would you like any discussion on this?

2. The notation of $\Delta l$ term is confused, why this term is out of the sumation over index i and j, since it is an entrywise calculation given specific i and j?

**Time Spent Reviewing:**

6 hours

---

> ### Author Response · Authors · 2021-08-10
> **Official response**
>
> Thank you for your review! Below we try to address each of your concerns; if any issues remain, we’d be very happy to discuss further.
>
> 1. This is another interesting way to relate SSL-HSIC to BYOL. In fact, InfoNCE can be decomposed and relates to BYOL in the same way you suggested; this is established in the BYOL paper, in Section 5. Also, an ablation study in the BYOL paper (also their Section 5) shows that adding the negative/contrastive terms back in the BYOL loss doesn't improve the results, and in some cases it hurts performance. This suggests that the learning mechanism of BYOL cannot be simply understood by the loss term used.
>
> 2. We also mentioned in our Section 3.2 that the relation to BYOL is more nuanced. The ways that predictor and target networks avoid trivial solutions are complex; the very recent paper https://arxiv.org/abs/2102.06810 demonstrates that the learning dynamics of BYOL-type losses are the result of multiple factors. To understand the dynamics, the authors had to resort to dynamical systems.
>
> 3. Regarding the empirical performance of SSL-HSIC, please see our separate comment to all reviewers.
>
> 4. The $\Delta l$ term reflects our special label structure: if two images come from the same original image $i$, then the distance between them,  $l(i, i)$, is a constant $l_1$ that is independent of $i$. If they come from two different images $i$ and $j$, then the distance between them is $l(i,j)=l_0$, also independent of the specific $i$ and $j$. As shown in Equation 13, $l(i,j)$ can be written as $\Delta l\\, \mathbb{I}(i=j) + l_0$. This allows $\Delta l$ to be pulled out of the summation. We explain this in more detail in Appendix A.1, but we’ll try to elaborate on this point in the main text too.

---

> > ### Comment · Reviewer_ArQY · 2021-08-23
> > **Thanks for your response**
> >
> > Thanks for your response on connection between BYOL, and I also appreciate your clarification on certain terms. However, I remain a bit concerned on your empirical evidence and the motivation of the paper. These are similar to the concerns raised by Reviewer Reviewer j3wZ. To me, the proposed method's novelty is centered at applying a new dependence metric on SSL problems, due to the reason "HSIC is extremely well understood" and "the reduced estimator bias of the corresponding method", along with its benefit in the "geometry" regard. The paper also shows close connections to various SSL method. However, your former response on the empirical study is still not fully convincing to me that HSIC is advantageous over MI in the regime of SSL methods, and therefore leaves the claim  "the good performance of SSL methods that use MI bounds can be explained by their closeness to HSIC, and not their relation to MI" questionable. I guess this is also partly because the section "Connection to InfoNCE" indeed proves certain connection (via many approximations) between HSIC and InfoNCE, but fails to deliver why HSIC is advantageous out of such provable connection? Still, stronger empirical study might help better advocate this point, I guess?

---

> > > ### Author Response · Authors · 2021-08-24
> > > **HSIC vs. MI clarification**
> > >
> > > Thank you for the response! We’d like to clarify that our contribution is a bit different. We claim that MI doesn’t explain SSL performance at all. Losses like InfoNCE perform well, but not because they are MI approximators.  In fact, MI is a poor criterion for representation learning.
> > >
> > > Here’s a simple explicit example where many possible representations have the same MI, but drastically different representation qualities. Consider a problem with two inputs, $A$ and $B$, and one-dimensional featurizer, parameterized by the integer $M$, which maps $A$ to $\mathrm{Uniform}(\{0, 2, \dots, 2M\}) + \mathrm{Uniform}((-0.5, 0.5))$ and $B$ to $\mathrm{Uniform}(\{1, 3, \dots, 2M+1\}) + \mathrm{Uniform}((-0.5, 0.5))$. When $M = 0$, the two inputs are simply separated into $A \in (-0.5, 0.5)$ and $B \in (0.5, 1.5)$. But if $M$ is large, they are interspersed like ABABABAB – a representation which is clearly much harder to work with for most downstream learners. Nevertheless, the mutual information between the features of any two augmentations of the same input (a positive pair) is independent of $M$, simply $H[Z_1] - H[Z_1 \mid Z_2] = \log 2$ for any $M$.
> > >
> > > Thus, to the extent that InfoNCE performs well, we argue that it is because it acts like HSIC. In turn, it follows that we don't expect very different empirical performance between HSIC and contrastive methods, but we do gain a better understanding of what contrastive methods are doing.

---

> > > > ### Comment · Reviewer_ArQY · 2021-08-24
> > > > **Thanks for your reply**
> > > >
> > > > Many thanks for your kind reply and nice illustration. I think your further explanation helps. I would like to maintain my current score then.

---

### Official Review · Reviewer_7VXB · 2021-07-16

**Rating:** 6
**Confidence:** 4

**Summary:**

This paper proposed a new Self-Supervised Learning framework with the Hilbert Schmidt Independence Criterion(SSL-HSIC). SSL-HSIC yields a new understanding of InfoNCE, while can optimize statistical dependence in time linear in the batch size.

**Limitations And Societal Impact:**

Although the paper gives detailed theoretical proof, the experiments are somewhat weak. I still have some concerns:
1）The most related works SwaV and Barlow Twins outperform the proposed method in some experimental results, as shown in Table 1,2,5. What are the main advantages of this method compared with SwaV and Barlow Twins?
2) HSIC(Z, Y) can be seen as a distance metric in the kernel space, where the cluster structure is defined by the identity. Although this paper maps identity labels into the kernel space, the information of one-hot label is somewhat limited compared with views embeddings in Barlow Twins.
3)Since the cluster structure is defined by the identity. How does the number of images impact the model performance? Do more training images make the performance worse or better ?

4) BYOL in the abstract should be explained for its first appearance.

**Main Review:**

This paper is well written with detailed proofs and theoretical results.
This paper gives a unified view of contrastive learning through dependence maximization, by establishing relationships between SSL-HSIC, InfoNCE, and metric learning with detailed analysis.

**Post Rebuttal Response**
I have read the response and would remain the current score.

**Time Spent Reviewing:**

4

---

> ### Author Response · Authors · 2021-08-10
> **Official response**
>
> Thank you for your review! Below we try to address each of your concerns; if any issues remain, we’d be very happy to discuss further.
>
> 1. Please see our comments to all the reviewers regarding the performance of SSL methods and the motivation of our paper.
>
> 2. Our one-hot “labels” use the same amount of information as the target covariance matrix in Barlow Twins, namely the knowledge of which images come from the same original one (positive examples). In our case, positive examples inform the “labels”; in Barlow Twins’ case, positive examples inform the target covariance matrix being an identity matrix.
>
> 3. The largest dataset we tried was ImageNet (about 1.3M images), so we can’t know for sure what would happen for even larger datasets. However, we’d expect larger datasets to require higher-dimensional features z so that separating many points becomes easier. (In Table 6d, we see that increasing the dimensionality of z increases the performance a bit.)
>
> 4. We will add the explanation of BYOL when it first appears in the abstract.

---

### Review · Ethics_Reviewer_CDAm · 2021-07-30

**Recommendation:** N/A

**Ethics Review:**

The paper describes a new independence criterion aimed at improving representation learning used for self-supervised learning. The paper was flagged for additional ethical review because of the lack of discussion of the ethical implications. Upon review, the paper does not appear to directly contain any potential negative societal impacts or issues of general ethical conduct.

---

### Review · Ethics_Reviewer_gL13 · 2021-08-11

**Recommendation:**

This proposed work calls for fuller ethical implication discussion to address: (1) how this proposed approach addresses which biases that encode, contain, or potentially exacerbate bias against people of a certain gender, race, sexuality, or who have other protected characteristics and (2) how ablation introduces, spreads and/or mitigates SSL-HSIC's representation quality. A much closer investigation of the randomization choices is warranted as SSL-HSIC's application shifts from animal ImageNet datasets to more real-world like face/person datasets. As a cascading effect of investigating these ethical implications, other more specific harms may be uncovered, such as but not limited to safety concerns and human rights concerns. An external body could be instrumental in reviewing how their proposed work would impact minoritized communities and pinpoint areas of racial/gender/class sanitization perpetuated by the algorithmic approach.

**Ethical Issues:**

Yes

**Ethics Review:**

The proposed Self-Supervised Learning with the Hilbert-Schmidt Independence Criterion (SSL-HSIC) directly transforms image data, which raises ethical concerns regarding both human rights infringement and bias encoding against people of a certain gender, race, sexuality, or who have other protected characteristics. As showcased on Lines 133-139, a series of data manipulations and encodings occur using randomization. The goal is to construct representations that are amenable for downstream tasks, e.g., scaling of selected similarity function.

---

> ### Author Response · Authors · 2021-08-24
> **Response to ethics review**
>
> Thank you for the review! We agree that the potential impacts of our work that you described are important and worth discussing. We currently discuss these issues concisely in Section 6 (Conclusions), but we will expand this discussion into a separate Broader Impact section.
>
> First, we would like to clarify the meaning of “bias” as we referred to it in Section 3.3. By bias of the HSIC estimator we mean statistical bias, not bias that is societal-oriented. A finite sample  estimator of a quantity is unbiased when it is equal to that quantity in expectation.
>
> Below is how we plan to expand our Broader Impact section so as to address your concerns in the review:
>
> > Our work concentrates on providing a more theoretically grounded and interpretable loss function for self-supervised learning. A better understanding of self-supervised learning dynamics, especially through more interpretable learning dynamics, is likely to lead to greater interpretability of the SSL framework generally, hence better and more explicit control over societal biases of these algorithms. SSL-HSIC yields an alternative, clearer understanding of existing self-supervised methods. As such, it is unlikely that our method introduces further biases than those already present for self-supervised learning.
>
> > The broader impacts of the self-supervised learning framework is an area that has not been studied by the AI ethics community, but we think it calls for closer inspection. An important concern for fairness of ML algorithms is dataset bias. ImageNet is known for a number of problems such as offensive annotations, non-visual concepts and lack of diversity, in particular for underrepresented groups. Existing works and remedies typically focus on label bias. Since self-supervised learning doesn’t use labels, however, the type and degree of bias could be very different from that of supervised learning. To mitigate the risk of dataset bias, one could employ dataset re-balancing to correct sampling bias [https://arxiv.org/pdf/1912.07726.pdf] or completely exclude human images from the dataset while achieving the same performance [https://openreview.net/pdf?id=BwzYI-KaHdr].
>
> > A new topic to investigate for self-supervised learning is how the bias/unbiased representation could be transferred to downstream tasks. We are not aware of any work in this direction.
> Another area of concern is security and robustness. Compared to supervised learning, self-supervised learning typically involves more intensive data augmentation such as color jittering, brightness adjustment, etc. There is some initial evidence suggesting self-supervised learning improves model robustness [https://arxiv.org/abs/1906.12340]. However, since data augmentation can either be beneficial [https://arxiv.org/pdf/1908.11229.pdf] or detrimental [https://www.aaai.org/AAAI21Papers/AAAI-353.YuD.pdf] depending on the type of adversarial attacks, more studies are needed to assess its role for self-supervised learning.
>
> Finally, we would like to emphasize (as we put it in the extended Broad Impact section) that we agree a thorough study of the social impact and ethics concerns of self-supervised learning is needed. Because the setting is fairly different from existing ethical analyses of which we’re aware, however, it would require a collaboration by authors across the field at large to prepare a specialized work covering SSL algorithms, and setting out appropriate experiments and metrics. New algorithmic papers could then use this framework to evaluate their new models.

---

### Author Response · Authors · 2021-08-10
**To all reviewers: a note on empirical performance and the motivation of our paper**

We would like to address the concerns about the performance of SSL-HSIC raised in several reviews by explaining our motivations and better contextualizing our results. We will improve the explanation provided in the paper as well.

Many contrastive methods for self-supervised learning are motivated by a mutual information optimization argument. The mutual information is problematic for several reasons:

- Mutual information is a notoriously challenging quantity to estimate accurately. Estimators with convergence guarantees are expensive, have significant bias, and suffer from a curse of dimensionality which makes them inaccurate in ML settings such as image data [see e.g. A, B]. In fact, at least in the discrete case no unbiased estimator exists [C, Proposition 8]. Meanwhile, currently-popular variational estimators satisfy few properties of the MI, suffer strong difficult-to-identify bias, and many have variance exponential in the value of the MI – a huge problem in high-MI regimes like those we’re trying to find in self-supervised learning [D, E].
- Even if we could estimate it, MI would not be the right criterion for self-supervised learning, since it ignores topology (see [33] for a detailed exposition; we also discuss this further, including a simple explicit illustration, at the start of [our response to Reviewer j3wZ](https://openreview.net/forum?id=0HW7A5YZjq7&noteId=2nbtZDBAWy1)).
- Self-supervised learning methods have been very successful with contrastive losses inspired by lower bounds on the MI. These bounds do respect topology (which is why the bounds are in practice “better” than the MI that inspired them), but their mathematical meaning has not been made clear -- they’re a poor MI approximation regardless of sample size [33, D, E], so “what do they converge to” as we see more samples? Without an understanding of the theoretical underpinnings of these criteria, how do we go about improving them?

Our work circumvents this problem:

- We show that a kernel divergence (HSIC) with a particular feature complexity penalty is in fact very close to the contrastive criteria that have been developed over the past few years. Unlike the approximate MI bounds discussed above, HSIC is extremely well understood. There is a simple population-level quantity that can be analyzed in terms of distributions. Moreover, there is an unbiased estimator, as well as the simpler slightly-biased estimator we use; both estimators converge to the population quantity at a rate of $1/\sqrt{n}$, and both estimators demonstrate topological properties consistent with the infinite sample limit. None of these properties are enjoyed by contrastive MI bounds. We believe that the good performance of SSL methods that use MI bounds can be explained by their closeness to HSIC, and not their relation to MI.
- In our empirical experiments, we verify that we get performance consistent with the current state-of-the-art: sometimes better, sometimes worse, depending on the particular task. This is further evidence that the properties of HSIC explain the performance of the related SSL losses. We would not expect radically better performance across the board, since our focus is rather on gaining a better and more principled understanding of contrastive losses as they are now used, through their strong relation with HSIC.

[A] S. Singh and B. Póczos (2016). Analysis of k nearest neighbor distances with application to entropy estimation. NeurIPS. https://arxiv.org/abs/1603.08578

[B] T. Berrett (2017). Modern k-Nearest Neighbour Methods in Entropy Estimation, Independence Testing and Classification. PhD Thesis, University of Cambridge. https://thomasberrett.github.io/thesis.pdf

[C] L. Paninski (2003). Estimation of Entropy and Mutual Information. Neural Computation. https://www.stat.berkeley.edu/~binyu/summer08/L2P2.pdf

[D] J. Song and S. Ermon (2020). Understanding the Limitations of Variational Mutual Information Estimators. ICLR. https://openreview.net/forum?id=B1x62TNtDS

[E] K. Stratos and D. McAllester (2020). Formal Limitations on the Measurement of Mutual Information. AISTATS. https://arxiv.org/abs/1811.04251

---

### Decision · Program_Chairs · 2021-09-27

**Decision:**

Accept (Poster)

**Comment:**

This paper studies the HSIC loss for self-supervised learning. While there was significant enthusiasm about this paper from the reviewers, there were also major concerns surrounding the conclusions of the experiments and questions about what to take away from the results. The authors do provide some good clarifications in their general response surrounding the fact that they don’t expect that HSIC should beat other methods and that this is not the point of the paper. However, the exact conclusions are still a bit muddy in the text.

An important point: there is a high degree of similarity between the objective underlying HSIC and BarlowTwins. This must be clarified in the paper. Now, the description is rather vague and there are no explicit ablations to compare or understand the differences between the two.